# Planning in Stochastic Environments with Awareness of Catastrophic Chance Events

## Abstract

Planning in stochastic environments is a research topic of high interest that remains underexplored in robustness, despite its importance for real-world applications. To consider robustness, previous methods typically assume an aggressive adversary that constantly attacks the agent, limiting the ability to learn and distorting the stochastic dynamics of chance events. In addition, expectation-centric methods implicitly discount low-probability events, failing to address rare catastrophes. To address this, we introduce *Robust Stochastic Zero*, the first method to be aware of catastrophes while maintaining the inherent stochastic dynamics. Specifically, it replaces the environment with a lurking adversary that mostly preserves the dynamics but selectively intervenes at the most critical moments. By targeting rarely occurring catastrophic chance events using tree-based planning, our method enables the agent to anticipate and avoid risky decisions, and also develops an adversary capable of delivering malicious impact with minimal intervention. On two benchmark stochastic environments, 2048 and Tetris Block Puzzle, Robust Stochastic Zero achieves an average of 122.1% of the baseline performance over both environments while intervening in only 0.05% of events, and remains comparable when no interventions occur. Our findings demonstrate that the right rather than constant intervention is a direction to robust planning in stochastic environments.

## 1 Introduction

Real-world decision-making is stochastic: *chance events* influence state transitions and uncertainty compounds over time. Planning in such environments requires balancing two objectives – maximizing expected returns and ensuring safety in extreme scenarios (Amodei et al., 2016; García & Fernández, 2015). Expectation-centric methods, which focus on maximizing expected returns, have yielded notable advancements (Antonoglou et al., 2021; Guei et al., 2022; Tesauro, 1995). However, they implicitly discount low-probability events and may fail to address catastrophes – extremely unfortunate events that occur only once along an otherwise safe trajectory comprising thousands of steps. As a result, agents that perform well on average may still make risky and vulnerable decisions precisely when catastrophes occur, limiting robustness and constraining deployment in safety-critical applications.

On the other hand, robust reinforcement learning methods address worst cases but are typically not tailored to stochastic catastrophes. Most of them (Pinto et al., 2017; Lecarpentier & Rachelson, 2019; Pattanaik et al., 2018; Zhang et al., 2020; Li et al., 2025) model the system as a minimax paradigm, where an adversary intervenes at every step, resulting in agent policies that are excessively conservative and even limiting the ability to learn. Frequent interventions also change the effective inherent stochastic dynamics, thereby distorting the problem the agent is meant to solve. The result is a conservative agent significantly not aligned with the inherent stochastic environment, yielding suboptimal average performance.

Motivated by this gap, we propose *Robust Stochastic Zero* (RSZ), the first method to be aware of rare catastrophes while maintaining the inherent dynamics in stochastic environments. RSZ departs from stepwise worst-case minimax modeling by letting an adversary intervene selectively at *critical afterstates* (the post-action, pre-chance intermediate), while leaving the inherent stochastic dynamics untouched elsewhere. Furthermore, RSZ is built upon Stochastic MuZero (Antonoglou et al., 2021),

a state-of-the-art method that leverages Monte Carlo tree search (MCTS) (Kocsis & Szepesvári, 2006) for tree-based planning in stochastic environments. By targeting critical afterstates in MCTS, RSZ contributes to robust planning in stochastic environments, enabling not only robust players who anticipate catastrophes and avoid risky decisions but also lurking adversaries who deliver malicious impact with minimal intervention. On two well-known stochastic games, 2048 and Tetris Block Puzzle, RSZ players achieve an average of 122.1% of the baseline performance over two games under interventions affecting only 0.05% of events; meanwhile, the players maintain comparable performance with no interventions. To summarize, our work demonstrates that making the right few interventions – not many – can deliver robust planning in stochastic environments.

## 2 STOCHASTIC ALPHAZERO AND STOCHASTIC MUZERO

*Stochastic AlphaZero* (SAZ) and *Stochastic MuZero* (SMZ) (Antonoglou et al., 2021) extend AlphaZero (Silver et al., 2018) and MuZero (Schrittwieser et al., 2020) to stochastic environments. Using the *afterstate* concept (Sutton & Barto, 1998), SAZ and SMZ integrate Monte Carlo tree search (MCTS) (Kocsis & Szepesvári, 2006; Coulom, 2007; Browne et al., 2012) with five key elements: *policy* $p$, *value* $v$, *chance event distribution* $\rho$, *afterstate value* $\nu$, and *reward* $r$ for planning.

**Planning** MCTS consists of three phases: *selection*, *expansion*, and *backup*. Initially, for an observation $o$ to be planned, the root node of the search is created. In selection, it selects recursively from the root node $s^0$ until reaching an unevaluated leaf node, which can either be a state $s^\ell$ or an afterstate $as^\ell$. At a non-leaf state $s^i$, the child node $as^{i+1}$ corresponds to the action $a^{i+1}$ that has the highest pUCT (Rosin, 2011; Silver et al., 2017) score is selected:

$$a^{i+1} = \arg\max_a \left[ q_a + p_a \times \frac{\sqrt{\sum_b n_b}}{n_a + 1} \times c_{\text{puct}} \right], \tag{1}$$

where $q_a$, $p_a$, and $n_a$ are the mean value, prior probability, and visit count for action $a$ at $s^i$, and $c_{\text{puct}}$ is a constant for exploration. Otherwise, at an afterstate $as^i$, the child node $s^{i+1}$ and its chance event $c^{i+1}$ with the highest *quasi-random sampling* (QRS) (Ozair et al., 2021) score is chosen:

$$c^{i+1} = \arg\max_c \left[ \frac{\rho_c}{n_c + 1} \right], \tag{2}$$

where $\rho_c$ is the true probability of taking chance event $c$ at $as^i$, and $n_c$ is the visit count. In expansion, the algorithm first executes the transition ($a^\ell$ or $c^\ell$) and obtains the reward $r^\ell$, where SAZ uses the perfect environment simulator while SMZ relies on its learned model. Next, the leaf node $s^\ell$ or $as^\ell$ is evaluated, obtaining $p^\ell, v^\ell$ or $\rho^\ell, \nu^\ell$, respectively. Then, the child nodes are expanded, using $p^\ell$ (or $\rho^\ell$) to initialize the prior probability (or true chance event probability). In backup, the mean values $q^i$ and the visit counts $n^i$ along the selection path are updated. At a selected state $s^i$, it updates

$$q^i \leftarrow \frac{n^i q^i + G^i}{n^i + 1} \text{ and } n^i \leftarrow n^i + 1, \tag{3}$$

where $G^i = \sum_{k=1}^{\ell-i} \gamma^{k-1} r^{k+i} + \gamma^{\ell-i} v^\ell$ sums the rewards $r^{i+1}, \ldots, r^\ell$ and the value $v^\ell$, discounted by a constant $\gamma$. At an afterstate $as^i$, the update follows the same pattern but replaces $v^\ell$ with $\nu^\ell$ in $G^i$. Repeating these phases yields the search policy $\pi$ at the root node for determining the action.

**Training** The algorithm collects self-play records that contain trajectories and planning results. As SAZ has access to the environment rules ($\rho$ and $r$ are directly available), its network predicts only $p$, $v$, and $\nu$, using a loss $L = \pi^\mathsf{T} \log p + (z - v)^2 + (z - \nu)^2 + c_{\text{norm}} ||\theta||^2$, where $z$ is the episode outcome (or n-step return), and the last term is an L2 regularization with a constant $c_{\text{norm}}$. On the other hand, SMZ plans with a learned model that predicts all of them. It also collects the observed reward $u$ and the happened chance event code $c$ to supervise $r$ and $\rho$, respectively.

## 3 ROBUST STOCHASTIC ZERO

*Robust Stochastic Zero* (RSZ) consists of a robust player and a lurking adversary, where the player anticipates catastrophes and avoids risky decisions while maintaining effectiveness in undistorted

dynamics, and the adversary delivers maximal impact with minimal interventions. The adversarial interventions are rare yet malicious, targeting only the most critical afterstates that significantly impact player performance, while permitting the less-critical afterstates to follow their inherent stochastic dynamics. To better anticipate catastrophes, RSZ utilizes MCTS to focus on the most critical afterstates where avoidable catastrophes are highly likely to occur.

## 3.1 AVOIDABLE CATASTROPHES

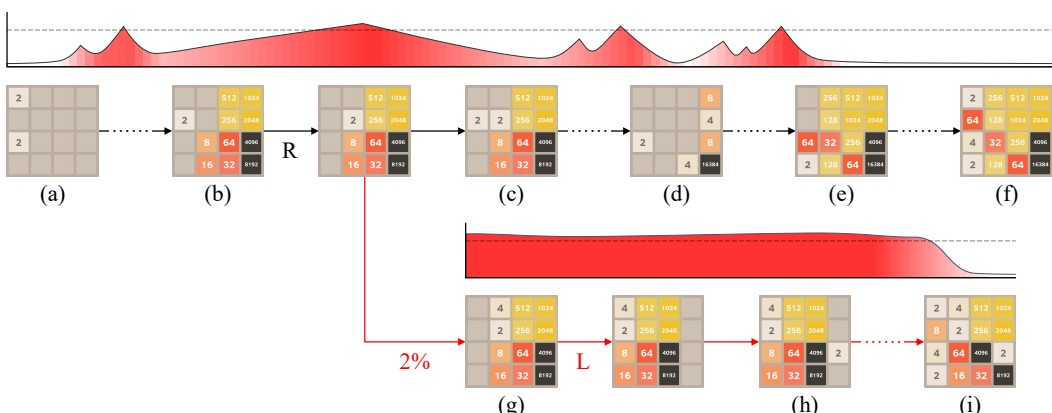

Figure 1: The avoidable catastrophes in 2048. Puzzles (a)-(f) show a normal game; (g)-(i) demonstrate an avoidable catastrophe after making a risky decision at (b). The chart above the puzzles measures the existence and the severity of such catastrophes.

Figure 1 illustrates the concept of avoidable catastrophes through the game-playing of 2048. The game begins with puzzle (a), where the player slides the puzzle to merge smaller tiles into larger tiles. After each move, a new tile is randomly placed in an empty location, and the game progresses. Later, in this example, the player encounters a puzzle (b), deciding to slide right and forming a Γ-shape empty space. Then, a 2-tile appears as in (c), and the game continues through (d), (e), and eventually ends at (f), with no moves remaining. However, the decision to slide right at (b) is risky: if a 4-tile is added with only a 2% chance at the specific location in (g), the player is forced to slide left. This shifts the largest tiles away from the border, making it possible for a subsequent unfortunate event to place a 2-tile like (h), which breaks the alignment. With extreme bad luck, a series of unfortunate events happens, and the game will end within four moves as shown in (i).

The *critical afterstates* are the points at which avoidable catastrophes can happen. Specifically, an afterstate is considered critical if and only if *some* of its chance events lead to catastrophes. For example, the situation becomes significantly critical following the risky decision at (b), as a 2% chance event (g) leads to a catastrophe (i), meanwhile (c) and others remain safe. Conversely, most situations in the game are not critical, as none of the chance events result in a relatively favorable or catastrophic outcome. Such as in (a) and (d), all chance events lead to similar results; while in (e), the game will end shortly regardless of the chance event.

As catastrophes happen with a low probability, expectation-centric methods can disregard them during training, resulting in the player being unaware of the risk and exposing vulnerabilities such as sliding right at (b). By targeting only critical afterstates, the adversary can concentrate its efforts to ensure that attacks are rare but highly malicious. Therefore, these infrequent attacks help the player learn to better avoid catastrophes from happening in an undistorted environment. Namely, when the player slides right at (b) during training, the adversary can attack as (g) and initiate the catastrophe, which ultimately helps the player learn to avoid such a risky decision.

## 3.2 ATTACKABILITY

*Attackability* is a measure of how critical an afterstate is, from the perspective of the antagonistic environment that wants to maximize the attack effectiveness and seeks to avoid frequent attacks.

The attackability $\tau$ of an afterstate $as$ is defined as

$$\tau = T(q, x)H(q, x), \tag{4}$$

where $q$ and $x = \{q_1, \ldots, q_n\}$ are the mean values of $as$ and its $n$ chance events, respectively. The design measures criticality considering both the severity $T$ and rarity $H$ of the potential catastrophes. For example, at Figure 1 (a), the puzzle has many empty locations with no larger tiles, so adding any tiles anywhere does not affect the final outcome; thus, (a) has low $T$ and low $H$. In contrast, at a vulnerable situation after sliding right at (b), an event (c) yields a significantly worse outcome with only a 2% chance – making both $T$ and $H$ very high. More cases are discussed in Appendix A.

**Measurement**    $T$ measures how much the player loses when the worst catastrophe happens, i.e., the value drop after taking the event with the minimum value:

$$T(q, x) = \text{clip}\Big(\frac{\tanh\big[c_T(-\frac{\min x}{q} + 1)\big]}{\tanh c_T}, 0, 1\Big), \tag{5}$$

where $c_T$ is a constant for adjusting the transformation. By design, $T$ increases as the worst event has a more negative impact on the mean value: $\min x = 0$ has the highest measurement, while $\min x = q$ has the lowest. On the other hand, $H$ indicates how uncommon the catastrophe is, i.e., how much the minimum event value stands out compared to other event values:

$$H(q, x) = \frac{\max \delta - \frac{1}{n}}{1 - \frac{1}{n}}, \text{ where } \delta = \text{softmax}\Big(\frac{q - x}{\sigma}\Big). \tag{6}$$

Here, $\delta = \{\delta_1, \ldots, \delta_n\}$ is the *value drop magnitudes*, measuring how much each event's mean value falls short of the parent value $q$, normalized by $x$'s standard deviation $\sigma$. The more $\delta$ approximates a one-hot vector, the more a single event stands out, and the better $H$ is considered. The detail of the measurement is provided in Appendix A.

**Attackable Threshold**    Given a *target attack rate* $\lambda$, we use the *attackable threshold* $\tau_{th}$ to consider the top $\lambda$ of the afterstates as critical. To prevent the attacks from being entirely predictable, the *attack adoption rate* $c_\lambda$ is also introduced. Namely, the adversary attacks when both $\tau > \tau_{th}$ and a chance of $c_\lambda$ are satisfied. However, a straightforward and fixed $\tau_{th}$ cannot adapt to players with varying decision-making preferences, as $\tau$ can exhibit diverse distributions and unpredictable bounds. An incorrect $\tau_{th}$ eventually results in overattacks or underattacks, negatively impacting performance. To address this, an estimator is introduced to dynamically determine $\tau_{th}$. Specifically, it estimates $\tau_{th}$ using observed $\tau$ of historical chance events such that exactly $\lambda$ of the afterstates satisfy $\tau > \tau_{th}$. The details and the implementation are provided in Appendix B.

### 3.3    Robust Stochastic Monte Carlo Tree Search

Robust Stochastic Monte Carlo tree search (RS-MCTS) enhances Monte Carlo tree search (MCTS) for robust planning in stochastic environments, incorporating attackability to target the most critical afterstates. It allows the adversary to initiate catastrophes through a series of strategic attacks, while also allowing the player to anticipate and avoid potential threats. The pseudocode is provided in Appendix B. The overview of the search is illustrated in Figure 2, where the three main phases – *selection*, *expansion*, and *backup* – are repeated throughout the planning, and the *decision* phase eventually determines the action to be taken.

**Selection**    The algorithm selects a trajectory from the root node $s^0$ (or $as^0$) to an unevaluated leaf node $s^\ell$ (or $as^\ell$). At each state $s^i$, as the player wants to maximize the outcomes, the next afterstate $as^{i+1}$ is selected by pUCT, exactly as in Equation 1. However, at each afterstate $as^i$, the lurking adversary may attack or keep the chance, depending on its attackability $\tau^i$, so the next state $s^{i+1}$ is selected by either pUCT or QRS, respectively. Specifically, if $\tau^i > \tau_{th}$, pUCT selects the child node using its value drop magnitudes as in Equation 1 with $\delta^i$ being the prior; otherwise, QRS selects the event using the true probabilities $\rho^i$ as in Equation 2. For a better determination, $\tau^i$ and $\delta^i$ are updated with the latest mean values using Equation 4.

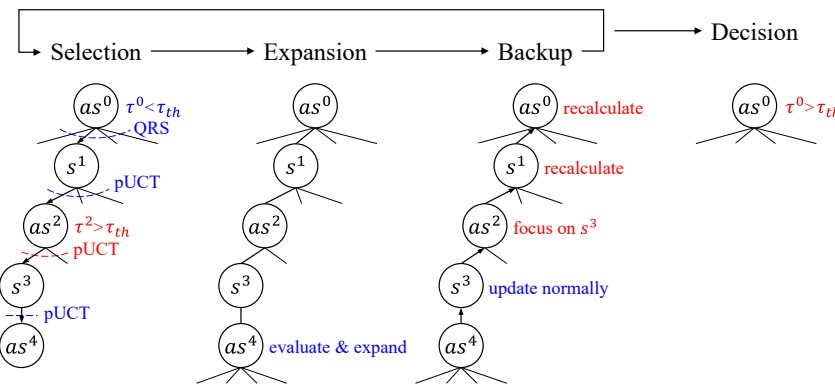

Figure 2: Robust Stochastic Monte Carlo tree search for planning in stochastic environments. In selection, $as^2$ has $\tau^2 > \tau_{th}$, so it selects $s^3$ using pUCT. In expansion, the leaf node $as^4$ is evaluated and expanded. In backup, $as^2$ is updated to focus on $s^3$'s mean value, and its ancestor nodes $s^1$ and $as^0$ are also updated to reflect the focus change. Finally, in decision, $as^0$ becomes attackable as a catastrophe has been anticipated by the search.

**Expansion**  The selected leaf node $s^\ell$ (or $as^\ell$) is evaluated by the neural network, after which its child nodes are expanded. This phase follows the original expansion phase in SAZ or SMZ, with added steps for a selected afterstate $as^\ell$: its value drop magnitudes $\delta^\ell = \{\delta_1^\ell, \dots, \delta_n^\ell\}$ are also retrieved from the network and assigned to each of its child nodes.

**Backup**  The newly evaluated value $v^\ell$ (or $\nu^\ell$) is used to update the statistics, $q^i$ and $n^i$, for all selected nodes. To capture the rare occurrence of attacks, we strengthen their outcomes by using a minimax design when attacks occur, enabling more efficient propagation from further lookahead to the root. Specifically, we update the mean value $q^i$ by using different rules depending on whether the node attacks:

$$
q^i \leftarrow \begin{cases} \dfrac{n^i(r^{i+1} + \gamma q^{i+1}) + \nu^i}{n^i + 1}, & \text{if the } i\text{th node attacks.} \quad (7) \\[2ex] \dfrac{\sum_a n_a(r_a + \gamma q_a) + v^i}{n^i + 1}, & \text{otherwise.} \quad (8) \end{cases}
$$

If the node attacks, namely for a node $as^i$ that selects its child node $s^{i+1}$ by pUCT, we apply Equation 7, where $r^{i+1}$ is the reward from $as^i$ to $s^{i+1}$, and $\nu^i$ is $as^i$'s afterstate value obtained through the neural network when $as^i$ was evaluated in its expansion. The formula strengthens the attack by updating the mean value to focus on the attacking event's mean value $q^{i+1}$, such that $q^i$ is updated as though the entire search budget so far was invested in $s^{i+1}$, enabling efficient propagation of the attack outcome. In addition, for applying this case, the selected node $s^{i+1}$ must be the worst-case event, namely the one whose value drop magnitude is the largest among $as^i$'s $\delta^i$. This ensures that $q^i$ can avoid focusing on less critical events visited by pUCT exploration.

Otherwise, the backup is essentially the same as SAZ or SMZ, which adds the newly evaluated value (with cumulative rewards) to the mean value $q^i$. However, if $\exists j > i$ such that $q^j$ has been updated using Equation 7, simply updating as in the original backup (Equation 3) cannot capture the focus change. Instead, for a state $s^i$, its $q^i$ is recalculated from all its child nodes by Equation 8, where $s^i$'s child node $a$ has visit count $n_a$, reward $r_a$, and mean value $q_a$. Note that for an afterstate $as^i$ that does not attack, Equation 8 still applies, but $v^i$ is replaced by $\nu^i$.

**Decision**  For the player's turn, an action is always determined based on MCTS statistics, adhering to the original SAZ or SMZ algorithm. However, for the environment's turn, the adversary may attack. Specifically, the chance event is determined as follows. If the root node $as^0$ satisfies $\tau^0 > \tau_{th}$, the attack proceeds with a chance of $c_\lambda$ by using MCTS statistics for decision. To prevent overattack, if the current attack rate is already higher than the target, the chance of attacking is further reduced, as detailed in Appendix B. Eventually, if no attack can be performed, a chance event is randomly chosen based on the true environment probabilities $\rho^0$.

### 3.4 TRAINING

For RSZ on top of SAZ, the neural network $\theta$ learns policy $p$, value $v$, afterstate value $\nu$, and value drop magnitudes $\delta$. The training of $p$, $v$, and $\nu$ follows the same process as those in the original SAZ, using the search policy $\pi$ and the outcome (n-step return) $z$ as the training targets. The training target for $\delta$ is denoted as $\zeta = \text{softmax}\left(\frac{\nu - x}{\sigma}\right)$, calculated from the parent afterstate value $\nu$ and complete chance event values $x$, normalized by $x$'s derivation $\sigma$. To obtain the complete $x$ for training, RS-MCTS ensures that an afterstate root node visits all its available chance events at least once. This design follows the behavior of QRS, which explores possible chance events one by one. Finally, the total loss combines the original loss in Section 2 and the value drop loss:

$$L = \pi^{\mathsf{T}} \log p + (z - v)^2 + (z - \nu)^2 + \zeta^{\mathsf{T}} \log \delta + c_{\text{norm}} ||\theta||^2. \tag{9}$$

## 4 EXPERIMENT

### 4.1 ROBUSTNESS ACROSS DIFFERENT LEVELS OF ATTACKS

To begin, we study the robustness of RSZ across different levels of evaluation-time attacks. For benchmarks, we take two popular single-player stochastic games, 2048 (Cirulli, 2014) and Tetris Block Puzzle (The Tetris Company, Inc, 2023), which are briefly introduced in Appendix C. The RSZ agents are trained with a desired attack rate $\lambda_{\text{train}} = 0.05\%$, and the SAZ agents are served as the baseline for comparison. The training and evaluation details are provided in Appendix D, and the training statistics are in Appendix E. At evaluation time, we vary $\lambda_{\text{test}} \in \{0\%, 0.05\%, \dots, 20\%\}$ and report RSZ and SAZ players' performance in Figure 3. Overall, under a setting of rare catastrophes happening with $\lambda_{\text{test}} = 0.05\%$, RSZ agents achieve an average performance of 122.1% of the baseline, with 122.9% and 121.3% in both games, respectively; and it also remains comparable in performance under environments without attacks ($\lambda_{\text{test}} = 0\%$). Such a finding aligns with our goal of being robust to rare but catastrophic chance events.

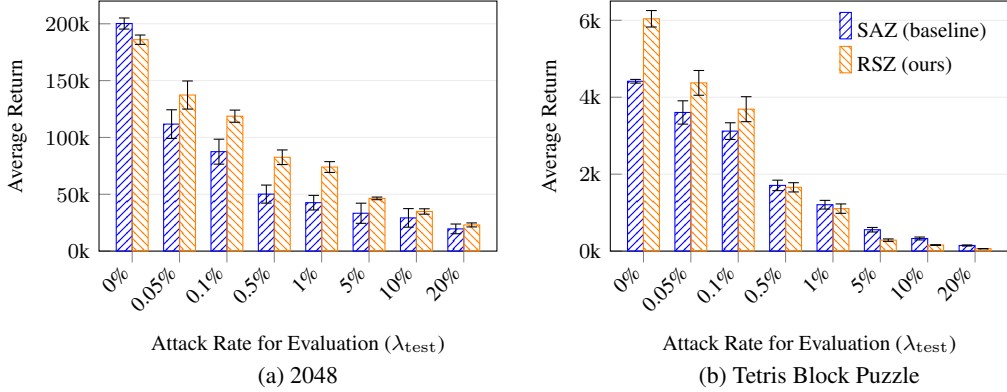

Figure 3: Player performance across attack rates. Higher is better.

In 2048, RSZ performs slightly behind but is still comparable to the baseline when $\lambda_{\text{test}} = 0\%$. A reason is that RSZ avoids gambling that depends on the specific next tile to trigger high-scoring merge chains, whereas SAZ is more willing to try such speculative outcomes – this may be required for this game. However, when $\lambda_{\text{test}} > 0\%$, RSZ consistently outperforms the baseline, indicating that its conservative planning better withstands adversarial attacks, while not losing too much performance when no attacking. In Tetris Block Puzzle, RSZ significantly outperforms the baseline when $\lambda_{\text{test}} = 0\%$. We hypothesize that, unlike 2048, its larger number of average available chance events (always 19) reduces the need for gambling, thus a conservative strategy produces a better result. In addition, RSZ maintains a clear advantage under small attack rates, e.g., $\lambda_{\text{test}} < 0.5\%$.

The evaluation suggests that for the rare catastrophe settings, $\lambda_{\text{test}} < 0.5\%$, RSZ trained with $\lambda_{\text{train}} = 0.05\%$ is capable of effectively handling them, demonstrating its scalability in adapting to $\lambda_{\text{test}}$ settings that differ from its training. In addition, considering that one of the goals of robust reinforcement learning is to maximize the lower bound, we also examine the return distribution

in both games, as detailed in Appendix F. The return distribution shows a significant difference between games. However, considering the performance at the 25th percentile, the RSZ players consistently show advantages over the SAZ players, indicating that robustness has been improved.

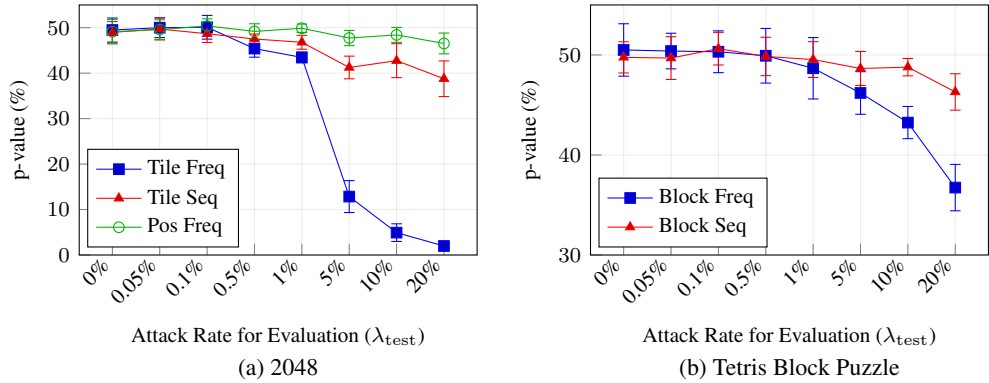

(a) 2048

(b) Tetris Block Puzzle

Figure 4: Randomness test across attack rates. Higher indicates better preservation of the inherent.

Following the evaluation of varying attack rates $\lambda_{\text{test}}$, we further analyze the distributional and sequential properties of chance events to verify whether the inherent randomness is preserved under different levels of attacks. The approaches are detailed in Appendix D. For 2048, we examine *tile frequency*, *tile sequence*, and *position frequency*, where tile frequency verifies the distribution for generating 2- and 4-tiles, tile sequence ensures the order of generated tile types is sufficiently random, and position frequency examines whether the new tiles are uniformly generated in empty locations. For Tetris Block Puzzle, we examine *block frequency* and *block sequence*, where block frequency determines whether new blocks are generated uniformly, and block sequence ensures that the block-generating orders are random enough. Figure 4 shows that the p-values remain high when $\lambda_{\text{test}} < 0.5\%$, confirming that RSZ maintains the inherent randomness with targeting rare catastrophes. On the other hand, the p-values significantly drop when $\lambda_{\text{test}} > 1\%$, demonstrating that frequent attacks distort the inherent randomness.

## 4.2 DEGRADATION UNDER TRAINING WITH MORE ATTACKS

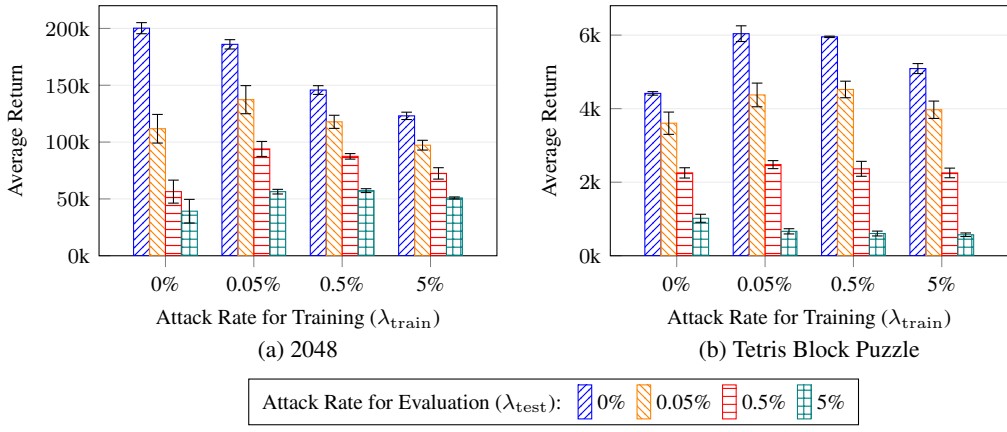

(a) 2048

(b) Tetris Block Puzzle

Attack Rate for Evaluation ($\lambda_{\text{test}}$): ▨ 0%  ▨ 0.05%  ▤ 0.5%  ▦ 5%

Figure 5: Comparison between agents trained with different attack rates. Higher is better.

To clarify whether frequent attacks used in most RARL methods distort the native stochastic dynamics and degrade performance, we train RSZ agents using higher attack rates and evaluate their performance. Specifically, we apply different evaluation-time attack rates $\lambda_{\text{test}} \in \{0\%, 0.05\%, 0.5\%, 5\%\}$ to test the agents trained with $\lambda_{\text{train}} \in \{0\%, 0.05\%, 0.5\%, 5\%\}$. Note that training with $\lambda_{\text{test}} = 0\%$ indicates the SAZ baseline. The comparison is summarized in Figure 5. First, when evaluation without attacks ($\lambda_{\text{test}} = 0\%$), in 2048, as the training-time attack rate $\lambda_{\text{train}}$ increases, the resulting agent

performs worse. In Tetris Block Puzzle, the performance reaches the best at $\lambda_{\text{train}} = 0.05\%$ and starts to decrease when $\lambda_{\text{train}} > 0.05\%$. Second, when evaluating with more attacks ($\lambda_{\text{test}} > 0\%$), there is no clear pattern showing that agents trained with a higher $\lambda_{\text{train}}$ improve or weaken their robustness. At the very least, these findings demonstrate that overly aggressive attacks can negatively impact performance in benchmark environments.

### 4.3 ABLATION ON STRAIGHTFORWARD METHODS

To better discriminate the effectiveness of RSZ, we further compare it with straightforward methods, including (1) **RSZ**: our method; (2) **RSZ-ns-worst-$v$**: the player runs RS-MCTS, and the adversary attacks by directly selecting the event with the worst value; (3) **RSZ-ns-$\delta$**: the adversary attacks by directly sampling an event from value drop magnitudes $\delta$; (4) **RSZ-n$\tau$**: the adversary attacks randomly without considering $\tau$; (5) **SAZ-$\tau$**: the player and the adversary run stochastic MCTS, while the adversary attacks with considering $\tau$; (6) **SAZ**: the baseline with no attack during training. Table 1 compares these methods by evaluating each method as a player and as an adversary, with the RSZ agent serving as the opponent in all evaluations. The results demonstrate that, compared with straightforward methods, RSZ can not only train a robust player but also train an effective adversary.

Table 1: Comparison between agents trained with RSZ and other straightforward methods in 2048. Higher is better when it acts as a player; lower is better when it acts as an adversary. Note that SAZ as adversary indicates that the RSZ players are performing in an environment without attacks.

| Method | As Player ($\uparrow$) | As Adversary ($\downarrow$) |
|---|---|---|
| **RSZ (ours)** | **137,318 $\pm$ 12,361** | **137,318 $\pm$ 12,361** |
| RSZ-ns-worst-$v$ | 135,453 $\pm$ 10,150 | 146,135 $\pm$ 10,651 |
| RSZ-ns-$\delta$ | 129,432 $\pm$ 8,075 | 151,250 $\pm$ 11,393 |
| RSZ-n$\tau$ | 127,988 $\pm$ 10,001 | 169,095 $\pm$ 15,837 |
| SAZ-$\tau$ | 133,534 $\pm$ 13,332 | 141,386 $\pm$ 6,762 |
| SAZ (baseline) | 111,742 $\pm$ 12,597 | 185,998 $\pm$ 4,114 |

### 4.4 BEHAVIOR ANALYSIS ON VULNERABILITY

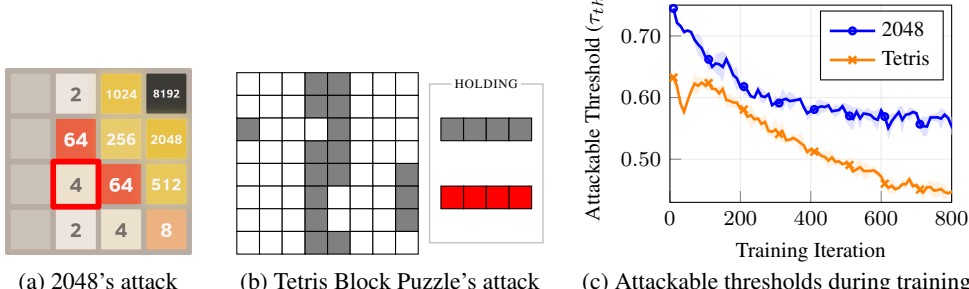

(a) 2048's attack      (b) Tetris Block Puzzle's attack      (c) Attackable thresholds during training

Figure 6: Examples of attacks and the changes in attackable thresholds during training. (a) and (b) are vulnerable afterstates with attacks highlighted; (c) shows attackable thresholds during training.

Finally, we analyze the vulnerability of decisions made by the RSZ agents during training. To achieve this, we first assess whether attackability $\tau$ can pinpoint the most critical afterstates. Figure 6 (a) and (b) illustrate afterstates we observed with high $\tau$ and their corresponding attacks. For (a), attacking with this 4-tile forces the player to slide the puzzle left, thereby initiating a series of attacks, similar to the catastrophe in Figure 1. For (b), the player must choose a block to place from the HOLDING panel; however, the added I-tetromino creates a situation where placement is impossible, resulting in an instant game over. More examples are available in Appendix G. From these examples, we conclude that the design of $\tau$ effectively targets vulnerable afterstates. Furthermore, we focus on the attackable threshold $\tau_{th}$, which is determined based on $\tau$. As afterstates result from the player's actions, a lower $\tau_{th}$ indicates that fewer afterstates present attractive attack targets,

showcasing improved player robustness. Figure 6 (c) tracks the changes of $\tau_{th}$ values during training. In both games, $\tau_{th}$ declines over time, demonstrating that RSZ increasingly improves policy toward safer afterstates and reduces the vulnerability exposed to the adversary.

## 5 RELATED WORKS

Robust decision-making is crucial for real-world applications and has thereby become a research topic of great interest in reinforcement learning. Notably, Pinto et al. (2017) proposed Robust Adversarial Reinforcement Learning (RARL) to model uncertainties as external disturbances through an adversary that applies destabilizing forces. In summary, prior works are typically not designed for stochastic environments (Tessler et al., 2019; Mandlekar et al., 2017; Pattanaik et al., 2018), using adversarial perturbations with added noise (Goodfellow et al., 2015; Gleave et al., 2019; Li et al., 2025), assume overly aggressive adversarial scenarios (Pinto et al., 2017; Lecarpentier & Rachelson, 2019), or overlook lookahead planning (Huang et al., 2022; Klima et al., 2019; Zhang et al., 2020). On the other hand, the critical state is a commonly used concept for improving reinforcement learning. Arjona-Medina et al. (2019) redistributed the final outcome to preceding states through return decomposition, and those states that receive a larger share are regarded as critical. Jacq et al. (2022) penalized the agent for choosing original actions instead of a random action and considered critical states to be those in which the agent persisted in original actions despite the penalty. Liu et al. (2023) identified critical states by jointly using a return predictor and a critical state detector, applying a soft mask over observed states to quantify their importance. Cheng et al. (2024) assigned each state a probability of taking a random action, interpreting lower probabilities as more critical since the agent prefers its learned policy over random actions in those states. In summary, these previous works identify a critical state either by return redistribution (Arjona-Medina et al., 2019) or by whether a random action can be taken (Jacq et al., 2022; Liu et al., 2023; Cheng et al., 2024).

Unlike prior RARL work, we plan in stochastic environments by leveraging the concept of critical afterstates. Our adversary intervenes selectively, attacking only critical afterstates, while most chance events still occur under the environment's inherent stochastic dynamics. This ensures that attacks will not distort the inherent dynamics. Additionally, for targeting critical afterstates, we measure the severity and rarity of potential catastrophes, which are computed from the value distribution over child nodes. In addition, we perform a tree-based planning for anticipating targeted attacks. Together, these design choices combine to create our approach that is both robust and effective in stochastic environments.

## 6 DISCUSSION

We introduce Robust Stochastic Zero (RSZ), the first tree-based planning method to be aware of rare catastrophes while preserving the inherent dynamics in stochastic environments. Across two benchmark environments, RSZ trains robust players that achieve 122.1% of the baseline performance in a catastrophe scenario, while maintaining comparable performance to the baseline in the absence of attacks. Also, it trains lurking adversaries that effectively deliver maximum malicious impact with minimal intervention of less than 1% of attacking events, while preserving the stochastic dynamics.

In this work, we implement RSZ on top of SAZ for computing efficiency, while it may also be applied to SMZ for robust model-based planning. However, a challenge is that implementing the lurking adversary requires the environment that interacts with the player to be modifiable. Furthermore, while this work focuses on single-player stochastic environments, extending RSZ to two- and even multi-player environments shows potential but requires further definition of the adversary's role and motives in player interactions. Overall, our work establishes a research direction for robust planning in stochastic environments built on sparse and well-timed interventions, potentially paving the way to the field of AI safety.

## REPRODUCIBILITY STATEMENT

The pseudocode of the proposed method is provided in Appendix B. The used benchmark environments are introduced in Appendix C. The training and evaluation specifications are detailed in

Appendix D. Upon acceptance, we will release the source code, trained models, configuration files, and scripts to reproduce all experiments.

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

# A DETAILS OF THE ATTACKABILITY

## A.1 EXAMPLES

Attackability is designed to use two parts, $T$ and $H$, as in Equation 4. By using $T$ and $H$ together, the measurement can take into account both the severity and rarity of the catastrophe. In Subsection 3.2, we discuss straightforward situations where $T$ and $H$ are both high or low. Here, we provide examples of when different $T$ and $H$ happen, involving five events chosen according to a uniform distribution.

**Consider Rarity Under Same Value Drop** Let two afterstates $as_1$ and $as_2$ have chance event values $x_1 = \{1, 3, 3, 4, 4\}$ and $x_2 = \{1, 1, 1, 6, 6\}$ with mean values $q_1 = 3$ and $q_2 = 3$, respectively. Now we calculate $T$ and $H$ for them, denote as $T_1$, $T_2$, $H_1$, and $H_2$. In this case, it is clear that $as_1$ should be considered more critical, as $as_1$ has only one worst-case event, while $as_2$ has three. For $T$, since the min value and the mean value are the same, their $T$ measures are also the same: $T_1 = T_2 = 0.77$ (assuming $c_T = 1$ in Equation 5 for simplicity). For $H$, the only worst-case event make $\max \delta$ in $H_1$ larger, resulting $H_1 = 0.61$ and $H_2 = 0.13$. So $as_1$ has a higher attackability.

**Consider Value Drop Under Same Rarity** On the other hand, let another two afterstates $as_3$ and $as_4$ have chance event values $x_3 = \{1, 3, 3, 3, 3\}$ and $x_4 = \{2, 3, 3, 3, 3\}$ with afterstate mean values $q_3 = 2.6$ and $q_4 = 2.8$, respectively. For this case, the rarity of their worst-case events appears similar, so we consider $as_3$ to be more critical, as its mean value drops by 1.6, which is larger than $as_4$'s 0.8. For $T$, the different min and mean values result in $T_3 = 0.72$ and $T_4 = 0.37$. For $H$, as we use standard deviation to normalize, the softmax outputs $\delta_3$ and $\delta_4$ are the same, resulting $H_3 = H_4 = 0.69$. This time, $as_3$ has a higher attackability.

## A.2 MEASUREMENT DETAILS

As in Equation 4, measuring attackability relies on $x$, which requires a complete enumeration of all chance events, making it less effective in practice. To address this, the attackability is measured using $\hat{x} = \{\hat{q}_1, \ldots, \hat{q}_n\}$, an estimation derived from the afterstate's mean value $q$, value drop magnitudes $\delta$, and two its visited chance events' mean values $q_a$ and $q_b$:

$$\hat{q}_i = q - \hat{\sigma}\left(\log \delta_i - \frac{1}{n}\sum_{\delta_j \in \delta} \log \delta_j\right), \text{ where } \hat{\sigma} = \max\left(\frac{q_b - q_a}{\log \delta_a - \log \delta_b}, 0\right). \tag{10}$$

Here, $q$, $q_a$, $q_b$, and $\delta$ are predicted by the neural network, during the RS-MCTS phases, as explained in Subsection 3.3. Note that measuring $\tau^i$ requires at least two visited chance events; otherwise, $\tau^i$ is defined as 0 in practice. However, the network can make inaccurate predictions. As $\delta$ presents the relative magnitudes of the value drops, $\delta_i$ is inversely related to $q_i$. The correct network prediction should yield a positive $\hat{\sigma}$ in Equation 10. However, when the neural network is not yet learned well, such as when $q_a > q_b$ and $\delta_a > \delta_b$, the calculation result turns negative. To address this, a $\max$ operator is introduced, allowing the estimation of $q_i$ to fail back to the parent afterstate value $q$.

## A.3 USING ATTACKABILITY IN MCTS

The following example illustrates how $\tau$ is calculated for an afterstate $as$ during tree search. First, when $as$ is first visited, its $\delta$ can be predicted by the neural network. The predicted $\delta$ allows the calculation of $H$ in Equation 6, but it is not sufficient for calculating $T$, resulting in $\tau = 0$. Then, after two of $as$'s child nodes $s_a$ and $s_b$ have been visited, their mean values $q_a = v_a$ and $q_b = v_b$ become available from the prediction network. At this stage, we calculate $\hat{x}$ using $\hat{q}_i$ of each node $s_i$ by Equation 10, and start measuring non-zero $T$ and $H$. As more child nodes are visited, we observed more accurate $q_i$ values for each child node. Therefore, we may update $\tau$ by replacing $\hat{q}_i$ with $q_i$ after each backup phase. Eventually, with sufficient visits, $\tau$ uses $x$, which is obtained completely with observed child node mean values $q^i$ without estimated $\hat{q}_i$.

# B  PSEUDOCODE AND ALGORITHM DETAILS

This section provides the pseudocode for procedures discussed in Subsection 3.2 and Subsection 3.3.

Algorithm 1 provides an overview of the Robust Stochastic Zero (RSZ) algorithm, with each functional component detailed in separate algorithms in the following paragraphs. Overall, Robust Stochastic Zero measures the attackability, identifies the most serious catastrophes with the threshold, and targets these most significant afterstates in the search to help the player avoid vulnerable decisions.

---

**Algorithm 1** Robust Stochastic Zero (RSZ)

---

**Parameter**: desired attack rate $\lambda^*$, search budget each move $n$
**Output**: network model parameters $\theta$
 1: $\theta \leftarrow$ initialize network parameters
 2: $\mathcal{T} \leftarrow$ initialize attack threshold estimator with $\lambda = \lambda^*$
 3: **while** training **do**
 4:     **while** collecting self-play records for a batch **do**
 5:         start a new episode
 6:         **while** the episode is not finished **do**
 7:             $\tau_{th} \leftarrow$ obtain latest threshold from $\mathcal{T}$
 8:             obtain the decision by RS-MCTS as in Algorithm 2
 9:             apply action or chance event transition
10:             update $\mathcal{T}$ using Algorithm 7 if it is a chance event
11:         **end while**
12:         add the finished episode into self-play record
13:     **end while**
14:     update $\mathcal{T}$ using Algorithm 8
15:     optimize network parameters $\theta$
16: **end while**
17: **return** $\theta$

---

This algorithm illustrates its training procedure; for evaluation, simply use the internal game-playing loop with a trained network $\theta$ and, optionally, an initial attackable threshold. Note that the algorithm may run multiple games concurrently to improve efficiency in practice.

Algorithm 2 shows an overview of the Robust Stochastic Monte Carlo tree search (RS-MCTS) algorithm, with each planning phase detailed later. Note that to initialize the search tree, the root node $s^0 \leftarrow h(o)$ is obtained from an observation $o$, using the *representation* network $h$ in SMZ. The final decision can be either an action or a chance event.

---

**Algorithm 2** Robust Stochastic Monte Carlo Tree Search (RS-MCTS)

---

**Input**: observation $o$, attackable threshold $\tau_{th}$, total search budget $n$
**Output**: the decision (action or chance event)
 1: initialize search tree
 2: **while** search budget available **do**
 3:     run selection phase using Algorithm 3
 4:     run expansion phase using Algorithm 4
 5:     run backup phase using Algorithm 5
 6: **end while**
 7: run decision phase using Algorithm 6
 8: **return** decision

---

Algorithm 3 shows the selection phase discussed in Section 3.3. For selection, both the root and leaf nodes can be a state or an afterstate; here we write $s^0$ and $s^\ell$ for simplicity. Note that $\tau^\ell$ and $\delta^\ell$ are always refreshed as in the pseudocode. However, there is no need to recalculate them in every selection in practice; one may simply update them when the mean value changes during backup.

---

**Algorithm 3** Selection in RS-MCTS

---

**Input**: root node $s^0$, attackable threshold $\tau_{th}$
**Output**: selected nodes $s^0, as^1, s^2, \ldots, s^\ell$

1: Select the root node $s^0$ and let $\ell \leftarrow 0$
2: **while** the last selected node is not a leaf node **do**
3:     **if** the last selected node is a state $s^\ell$ **then**
4:         $as^{\ell+1} \leftarrow$ select a child node from $s^\ell$ by pUCT as in Equation 1
5:         Select the subsequent node $as^{\ell+1}$ and let $\ell \leftarrow \ell + 1$
6:     **else if** the last selected node is an afterstate $as^\ell$ **then**
7:         $\tau^\ell \leftarrow$ calculate the attackability of $as^\ell$ by Equation 4
8:         **if** $\tau^\ell > \tau_{th}$ **then**
9:             $\delta^\ell \leftarrow$ calculate value drop magnitudes of $as^\ell$ as in Equation 6
10:             $s^{\ell+1} \leftarrow$ select a child node from $as^\ell$ by pUCT as in Equation 1 (using $\delta^\ell$ as the prior)
11:         **else**
12:             $s^{\ell+1} \leftarrow$ select a child node from $as^\ell$ by QRS as in Equation 2
13:         **end if**
14:         Select the subsequent node $s^{\ell+1}$ and let $\ell \leftarrow \ell + 1$
15:     **end if**
16: **end while**
17: **return** all selected nodes $s^0, as^1, s^2, \ldots, s^\ell$

---

Algorithm 4 shows the expansion phase discussed in Section 3.3. In this phase, five model functions are used: the *prediction* $f$, *afterstate dynamics* $\phi$, *afterstate prediction* $\psi$, *dynamics* $g$, and *value drop magnitudes prediction* $\kappa$. These functions utilize a learned neural network in SMZ. However, in SAZ, since the perfect environment simulator directly provides $g$, $\phi$, and $\rho$ in $\psi$, only the other parts require a neural network.

---

**Algorithm 4** Expansion in RS-MCTS

---

**Input**: leaf node $s^\ell$ (or $as^\ell$)
**Parameter**: neural network parameters $\theta$
**Output**: leaf node value $v^\ell$ (or $\nu^\ell$), action reward $r^\ell$ if available

1: **if** the leaf node is a state $s^\ell$ **then**
2:     $s^\ell, r^\ell \leftarrow g(as^{\ell-1}, c^\ell)$                    ▷ the *dynamics* network
3:     $p^\ell, v^\ell \leftarrow f(s^\ell)$                    ▷ the *prediction* network
4:     **for** $s^\ell$'s all legal player actions $a$ **do**
5:         Expand its child node $a$
6:         Set its prior as $p_a^\ell \in p^\ell$
7:     **end for**
8: **else if** the leaf node is an afterstate $as^\ell$ **then**
9:     $as^\ell \leftarrow \phi(s^{\ell-1}, a^\ell)$                    ▷ the *afterstate dynamics* network
10:     $\rho^\ell, \nu^\ell \leftarrow \psi(as^\ell)$                    ▷ the *afterstate prediction* network
11:     $\delta^\ell \leftarrow \kappa(as^\ell)$                    ▷ the *value drop magnitudes prediction* network
12:     **for** $as^\ell$'s all possible chance events $c$ **do**
13:         Expand its child node $c$
14:         Set its value drop magnitude and event probability as $\delta_c^\ell \in \delta^\ell$ and $\rho_c^\ell \in \rho^\ell$, respectively
15:     **end for**
16: **end if**
17: **return** $v^\ell$ (or $\nu^\ell$) and $r^\ell$ if available

---

Algorithm 5 shows the backup phase discussed in Section 3.3. Note that both the root and leaf nodes in the selection path can be a state or an afterstate; here we write $s^0$ and $s^\ell$ for simplicity.

---

**Algorithm 5** Backup in RS-MCTS

---

**Input**: selected nodes $s^0, as^1, \ldots, s^\ell$, value $v^\ell$ (or $\nu^\ell$), reward $r^\ell$

1: Set value $v^\ell$ for $s^\ell$
2: Set reward $r^\ell$ for the transaction from $as^{\ell-1}$ to $s^\ell$
3: **for** $s^i$ (or $as^i$) **in** $s^\ell, as^{\ell-1}, \ldots, s^0$ **do**
4:    **if** $as^i$ is attackable **and** $s^{i+1}$ is the highest in $\delta^i$ **then**
5:       Update $q^i$ and $n^i$ by focusing on the next selected state, as in Equation 7
6:    **else**
7:       Update $q^i$ and $n^i$ by calculating mean value from all its child node, as in Equation 8
8:    **end if**
9:    Update normalization bound with updated $q^i$
10: **end for**

---

Algorithm 6 shows the decision phase discussed in Section 3.3. Here, the attack rate $\lambda$ in the input list is a ground truth attack rate ($\lambda_{\text{train}}$ or $\lambda_{\text{test}}$) that we ultimately want. $\hat{\lambda}$ is the attack rate observed in the current game, starting from the initial state to the current time step. In practice, when multiple games are running in parallel, $\tau_{\max}$ is shared among them.

---

**Algorithm 6** Decision in RS-MCTS

---

**Input**: root node $s^0$ (or $as^0$), attackable threshold $\tau_{th}$, attack rate $\lambda$
**Output**: the decision (action or chance event)

1: **if** the root node is a state $s^0$ **then**
2:    **return** an action chosen by MCTS
3: **else if** the root node is an afterstate $as^0$ **then**
4:    $\tau \leftarrow$ calculate attackability score of $as^0$ by Equation 4
5:    **if** $\tau > \tau_{th}$ **then**
6:       $\hat{c}_\lambda \leftarrow c_\lambda$                    $\triangleright$ the attack adoption rate
7:       $\tau_{\max} \leftarrow \max(\tau_{\max}, \tau)$
8:       $\hat{\lambda} \leftarrow$ calculate attack rate of current game
9:       **if** $\hat{\lambda} \geq \lambda$ **then**
10:         $\hat{c}_\lambda \leftarrow \hat{c}_\lambda \left(\frac{\tau - \tau_{th}}{\tau_{\max} - \tau_{th}}\right)^2$         $\triangleright$ the chance to overattack
11:       **end if**
12:    **else**
13:       $\hat{c}_\lambda \leftarrow 0$
14:    **end if**
15:    **if** hit a chance of $\hat{c}_\lambda$ **then**
16:       **return** a chance event chosen by MCTS
17:    **else**
18:       **return** a chance event chosen randomly
19:    **end if**
20: **end if**

---

Algorithm 7 shows the attackable threshold estimator, as discussed in Section 3.2. The estimator observes the attackability $\tau$ of chance events that happen, updates $\tau_{th}$ by $\tau$. Essentially, it is an online quantile estimator, modified from the *Fast Algorithm for Median Estimation* (FAME) (Feldman & Shavitt, 2007). In practice, as multiple concurrently running games can contribute to a single instance of the estimator, it can quickly estimate $\tau_{th}$ for targeting critical afterstates.

---

**Algorithm 7** Attackable Threshold Estimator

---

**Input**: observed attackability score $\tau$
**Parameter**: attackable threshold $\tau_{th}$, attack rate $\lambda$, step size $k$
**Output**: updated attackable threshold $\tau_{th}$
 1: **if** $\tau_{th} > \tau$ **then**
 2:   $\tau_{th} \leftarrow \tau_{th} - \lambda k$
 3: **else if** $\tau_{th} < \tau$ **then**
 4:   $\tau_{th} \leftarrow \tau_{th} + (1 - \lambda)k$
 5: **end if**
 6: **if** $|\tau_{th} - \tau| < k$ **then**
 7:   $k \leftarrow 0.5k$
 8: **end if**
 9: **return** $\tau_{th}$

---

However, accurately targeting the threshold can be challenging since the distribution of attackability scores is often highly imbalanced, varies from player to player, and can shift during training when the player's behavior changes rapidly. For example, in an extreme scenario where 99% and 1% of observed $\tau$ are 0s and 1s respectively, there is no effective threshold to exactly target the top 0.05% critical afterstates.

Algorithm 8 shows the target controller that works together with the estimator to further improve the accuracy of $\tau_{th}$. It operates periodically, monitoring whether the attack rate $\hat{\lambda}$ in recent decision history is significantly off the desired target attack rate $\lambda^*$. If this happens, the estimator's target $\lambda$ is adjusted, allowing the estimator to fit a more relaxed target. Note that $\lambda^*$ is a fixed target (like $\lambda_{\text{train}}$ or $\lambda_{\text{test}}$ in experiments), $\lambda$ is a varying target used by the estimator, and $\hat{\lambda}$ is observed from self-play records. For example, if the error is $\lambda^* - \hat{\lambda} = 0.005$ when $c_\lambda = 0.5$, the adjustment will be $\lambda \leftarrow \lambda + \frac{0.005}{0.5}$, triggering more attacks to reduce the error.

---

**Algorithm 8** Target Controller for Attackable Threshold Estimator

---

**Input**: self-play records $D$ of the current iteration
**Parameter**: target attack rate $\lambda^*$, attack rate $\lambda$, step size $k$
 1: $\hat{\lambda} \leftarrow$ calculate average attack rate in $D$
 2: **if** $|\lambda^* - \hat{\lambda}| \geq 0.05\lambda^*$ **then**
 3:   **if** $k$ was just been reset in the last iteration **then**
 4:      Adjust attack rate: $\lambda \leftarrow \text{clip}\big(\lambda + \frac{\lambda^* - \hat{\lambda}}{c_\lambda}, 0, 1\big)$
 5:   **end if**
 6:   Reset step size: $k \leftarrow 0.2$
 7: **end if**

---

## C  2048 AND TETRIS BLOCK PUZZLE

**2048** (Cirulli, 2014) is a single-player, perfect information stochastic game. 2048 is played on a $4 \times 4$ puzzle, where all tiles are numbered in powers of two. At each turn, the player can slide the puzzle left, right, up, or down to move all tiles as far as possible in the chosen direction. When two $k$-tiles collide during a slide, they are merged into a single $2k$-tile, and the player receives a reward of $2k$. For example, two 2-tiles can be merged into a single 4-tile, receiving 4 points. After each move, the environment uniformly selects an empty location to generate a new 2-tile or 4-tile with probabilities 90% and 10%, respectively. The goal of 2048 is to achieve the 2048-tile and accumulate as many points as possible until no further moves are available.

**Tetris Block Puzzle** (The Tetris Company, Inc, 2023) is also a single-player, perfect information stochastic game. In this work, we use a slightly modified version. The game is played on an $8 \times 8$ board, and starts with two holding blocks. Each turn, the player selects one tetromino from the holding blocks and places it on the board without any rotation. When a line (a row or a column) is filled, it will be eliminated. After placement, the player receives a reward of $n^2$ points, where $n$ is the number of the eliminated lines. Also, a new holding block will be randomly added from the 19 possible Tetris tetrominoes (7 base shapes, including all valid rotations). The game ends when there are no more spaces to place the tetromino, or when a maximum of 13,500 steps has been reached.

## D  EXPERIMENT DETAILS

### D.1  TRAINING

In this section, we describe the details for training Robust Stochastic Zero (RSZ) models. Our RSZ implementation is built upon MiniZero (Wu et al., 2025), a publicly available AlphaZero/MuZero framework. In addition, we implement it on top of Stochastic AlphaZero (SAZ) (Antonoglou et al., 2021) and Gumbel AlphaZero (GAZ) (Danihelka et al., 2022), to improve training efficiency. For the training configurations, we generally follow those in SAZ, where the hyperparameters are listed in Table 2.

Table 2: Hyperparameters for training.

| Parameter | 2048 | Tetris |
|---|---|---|
| Optimizer | Adam | |
| Optimizer: learning rate | 0.001 | |
| Optimizer: weight decay | 0 | |
| Discount factor | 0.999 | |
| Priority exponent ($\alpha$) | 0.6 | |
| Priority correction ($\beta$) | 0.4 | |
| Bootstrap step (n-step return) | 1 | |
| # Self-play actors | 2048 | |
| MCTS simulation | 50 | |
| Softmax temperature | 1 | |
| Iteration | 800 | |
| Training steps | 51,200 | |
| Batch size | 1024 | |
| # Blocks | 3 | |
| Replay buffer size | 50,000 positions | |
| Attack adoption rate ($c_\lambda$) | 0.5 | |
| Attackability constant ($c_T$) | 10 | |
| Gumbel $c_{\text{visit}}$ | 50 | |
| Gumbel $c_{\text{scale}}$ | 1 | |
| Gumbel sample actions ($m$) | 3 | 12 |

The experiments are conducted on the machine equipped with two E5-2678 v3 CPUs and four GTX 1080 Ti GPUs. For each experiment setting, we train five network models with different seeds; each model is trained for 800 iterations with a total of 51,200 training steps. For 2048, each model takes

approximately 25 hours to complete. Specifically, for $\lambda_{\text{train}} = 0.05\%$, 0.5%, and 5%, the training times are approximately 24.83, 21.44, and 20.04 hours, respectively. This speedup is caused by the average game lengths under a more aggressive adversary being shorter. For Tetris Block Puzzle, each model takes approximately 36 hours to complete. Specifically, for $\lambda_{\text{train}} = 0.05\%$, 0.5%, and 5%, the training times are approximately 36.30, 29.76, and 23.22 hours, respectively. Like 2048, this speedup is caused by more attacks.

## D.2    EVALUATION

To ensure statistical stability when evaluating the performance, after training five models for each setting, we pre-evaluate them using $\lambda_{\text{test}} = 0\%$. Each evaluation plays 200 games to obtain a stable average return. Then, we remove the two models that perform the best and the worst during the pre-evaluation, thereby resulting in three remaining models for the main evaluations and analyses.

For the evaluations with $\lambda_{\text{test}} = 0\%$, the performance is averaged over the three player models, using their pre-evaluation statistics. For the evaluations with $\lambda_{\text{test}} > 0\%$, to evaluate the performance between a player setting and an adversary setting, we pair-match the player and adversary models, conducting $3 \times 3$ player-vs-adversary evaluations. During these evaluations, all the adversaries use their own attackable threshold $\tau_{th}$ as an initial $\tau_{th}$ to make it easier for the adversaries to find a reasonable threshold, rather than adjust it from scratch. Note that if we have the historical records of the player, a better initial $\tau_{th}$ can be calculated from them, better reflecting the decision preference.

## D.3    ANALYSIS

In the randomness test, we calculate the average p-value over 200 games for each model, and then calculate the average and standard deviation of the p-values across three models. The p-value indicates whether the trajectories pass the randomness test. For 2048, tile frequency is assessed with the Chi-square test (van der Vaart, 1998) against the distribution of tile type. Tile sequence uses the runs test (Bradley, 1960) to ensure the order of generated tile types is sufficiently random For example, if a 4-tile is always generated after nine 2-tiles, it satisfies the probability rule but is very predictable. Position frequency applies the Chi-square test to examine whether the new tiles are uniformly generated in empty locations. For Tetris Block Puzzle, block frequency is checked with the Chi-square test to examine whether the new blocks are generated uniformly. Block sequence uses the serial test (Knuth, 1997) to ensure the block-generating sequences are random enough.

In the vulnerability analysis, we calculate the average and standard deviation of the attackable thresholds of three robust players during the training process, using their self-play trajectories.

# E   TRAINING STATISTICS

## E.1   AVERAGE RETURNS

We track the average return of SAZ and RSZ agents during training. The training curves are shown in Figure 7, where the data is averaged from five trained models for each method. Note that SAZ is trained without any adversary's attack, while RSZ is trained under adversary's attacks with an attack rate $\lambda_{\text{train}} = 0.05\%$. From the curves, RSZ agents perform worse than SAZ agents consistently during training in 2048. Nevertheless, RSZ agents perform better most of the time in Tetris Block Puzzle, showing that the attacks help players avoid creating unfavorable situations.

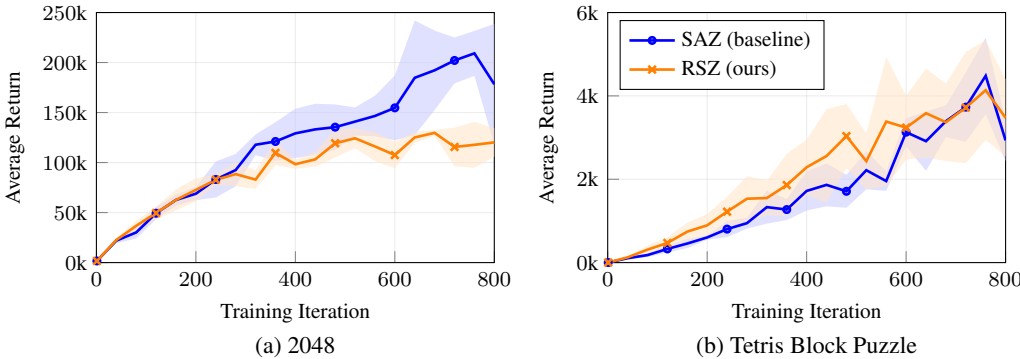

(a) 2048                                    (b) Tetris Block Puzzle

Figure 7: Average returns during training. Higher is better. Note that SAZ is trained without attacks, while RSZ is trained with an attack rate of 0.05%.

## E.2   AVERAGE ATTACK RATE

We also track the attack rate of RSZ agents during training, as shown in Figure 8. As shown in the figure, the attack rate in 2048 remains quite stable during training. On the other hand, the attack rate in Tetris Block Puzzle exhibits some instability, likely due to situations discussed in Appendix A. Since the attackable threshold cannot be estimated precisely, overattacks happen.

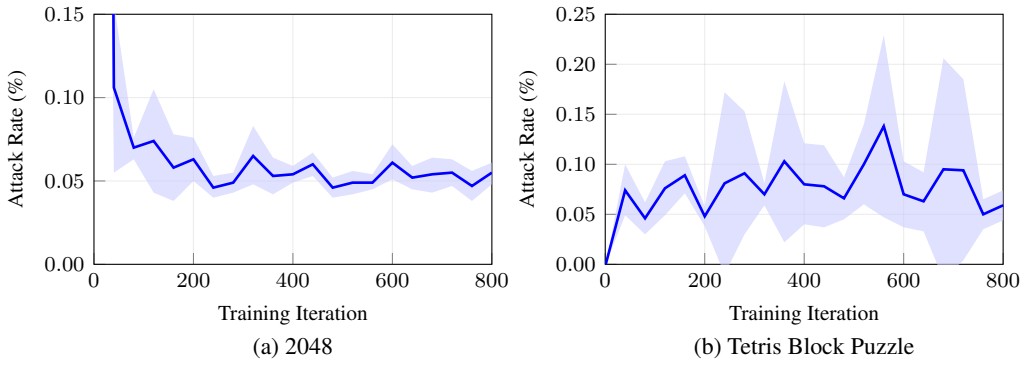

(a) 2048                                    (b) Tetris Block Puzzle

Figure 8: Attack rates observed during training. Target attack rate is 0.05%.

# F EVALUATION STATISTICS

## F.1 RETURN DISTRIBUTION

Since one of the targets of robust reinforcement learning is to maximize the lower bound, we also use box plots to visualize the results in Figure 3, which better reflects the data distribution. Figure 9 shows the results of the RSZ and SAZ players in evaluations with and without rare catastrophes ($\lambda_{\text{test}} = 0\%$ and $0.005\%$). The results demonstrate that the RSZ players have a narrowed return distribution in 2048, but the situation is completely the opposite in Tetris Block Puzzle.

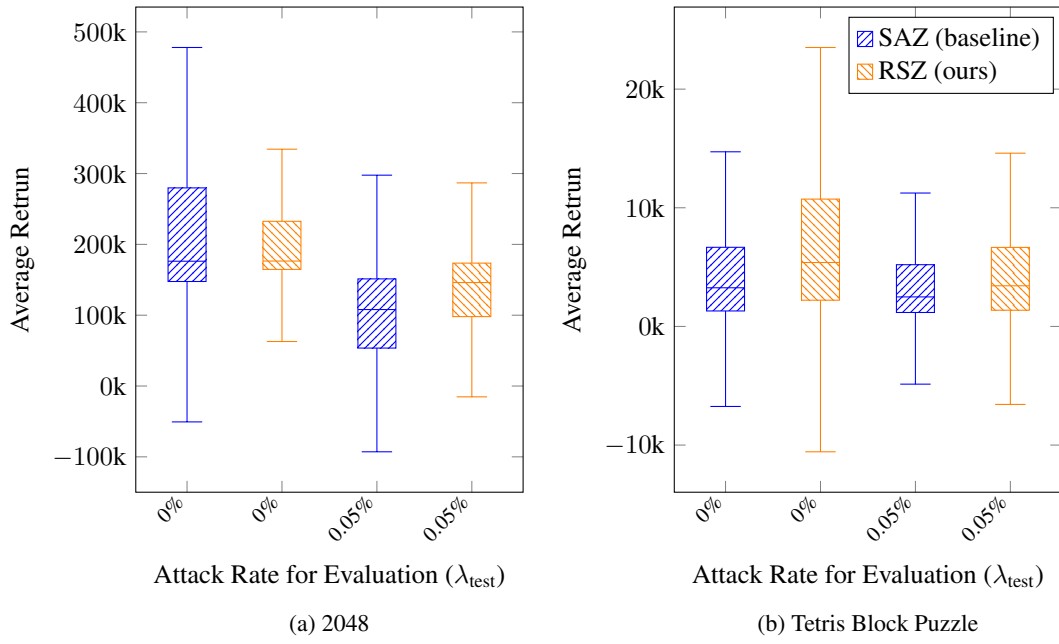

(a) 2048

(b) Tetris Block Puzzle

Figure 9: Box plot for environments with and without attacks ($\lambda_{\text{test}} = 0\%$ and $0.05\%$).

Furthermore, for the detailed distribution of the average return, we also investigate the returns of each episode during evaluations, and visualize them using the percentile plot, as shown in Figure 10 and Figure 11. Although RSZ performs worse than SAZ in the best games, it performs better than the baseline in the worst situations, demonstrating its robustness in withstanding catastrophic chance events.

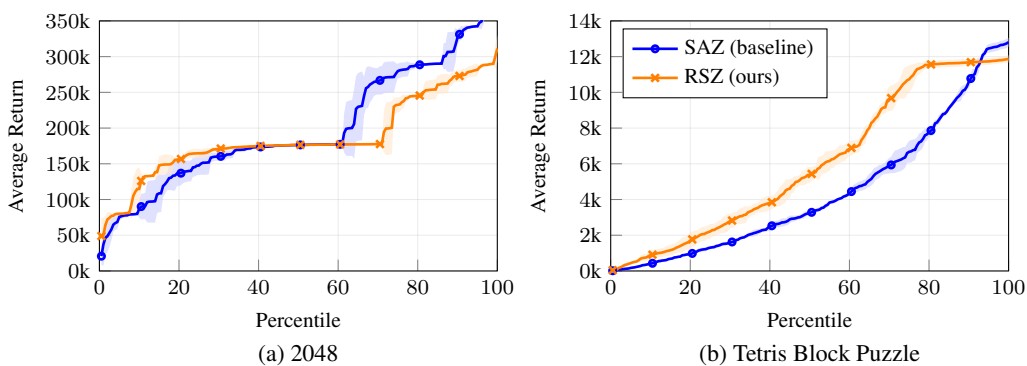

(a) 2048

(b) Tetris Block Puzzle

Figure 10: Percentile plots for the episode returns without attacks. Higher is better.

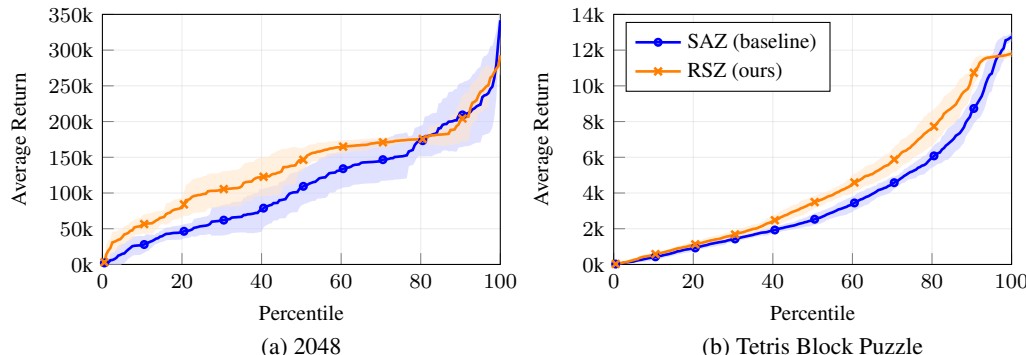

(a) 2048

(b) Tetris Block Puzzle

Figure 11: Percentile plots for the episode returns under an attack rate of $\lambda_{\text{test}} = 0.05\%$. Higher is better.

## G   MORE BEHAVIOR ANALYSIS

### G.1   EXAMPLES OF VULNERABLE AFTERSTATES

We gather critical and non-critical afterstates from self-play records in both games, as shown in Figure 12 and Figure 13. The critical afterstates are targeted by the adversary, with the attacking events highlighted in the figure. Due to their high attackability scores, these attacks either cause an immediate game over or initiate a chain of events that eventually lead to a game over. On the other hand, afterstates with low attackability show no difference between chance events, so they are not favored targets for attack. This underscores the effectiveness of the attackability in identifying critical afterstates.

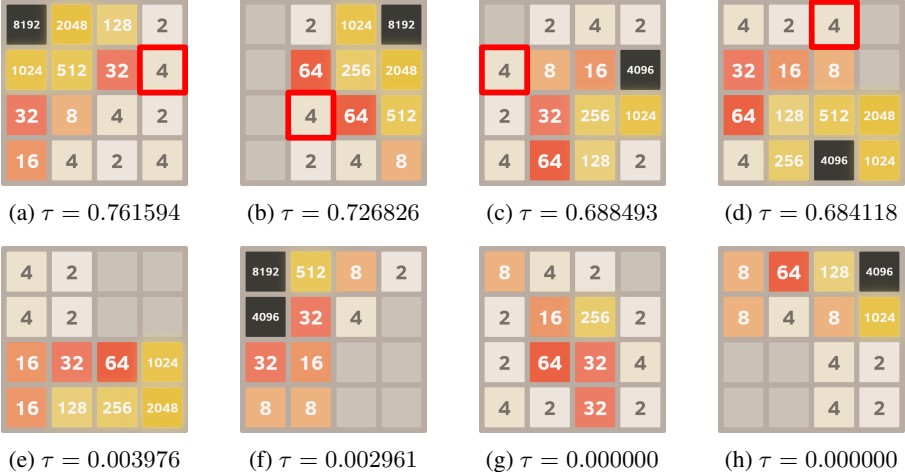

Figure 12: Vulnerable and invulnerable afterstates in 2048. (a)-(d) are vulnerable afterstates with their corresponding attacking event shown as the highlighted tile (not being placed on the afterstate when calculating $\tau$); (e)-(h) are invulnerable afterstates.

### G.2   ATTACKABILITY DISTRIBUTION

To gain better insights into how decision vulnerability evolves at different stages of training, we analyze the attackability distribution from self-play records across all trained RSZ agents, as plotted in Figure 14 and Figure 15. At the 1st iteration, the distributions skew towards higher attackability, indicating that the untrained agents make vulnerable decisions. By the 800th iteration, the distribution shifts markedly toward lower attackability, evidencing that training progressively improves the policy toward more robust decisions.

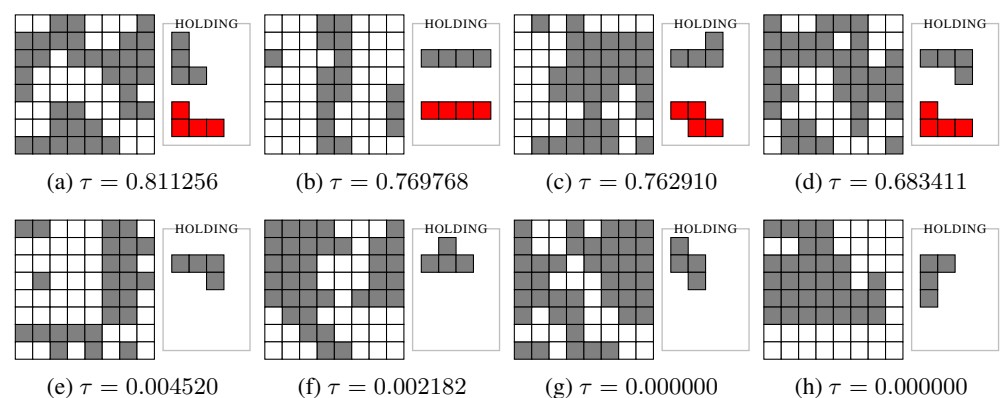

(a) $\tau = 0.811256$    (b) $\tau = 0.769768$    (c) $\tau = 0.762910$    (d) $\tau = 0.683411$

(e) $\tau = 0.004520$    (f) $\tau = 0.002182$    (g) $\tau = 0.000000$    (h) $\tau = 0.000000$

Figure 13: Vulnerable and invulnerable afterstates in Tetris Block Puzzle. The blocks to the right of each puzzle are holding blocks. (a)-(d) are vulnerable afterstates with their corresponding attacking event shown as the highlighted holding block (not being placed on the afterstate when calculating $\tau$); (e)-(h) are invulnerable afterstates. Note that in an afterstate, there is only one holding block, as another has just been placed on the puzzle.

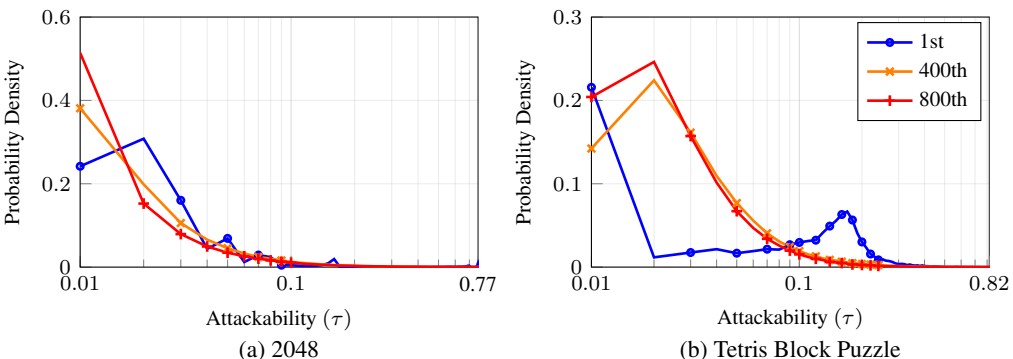

(a) 2048              (b) Tetris Block Puzzle

Figure 14: Probability density of attackability during training. The minimum $\tau$ is 0, displayed together with 0.01. The maximum $\tau$ is 0.77 and 0.82 in two games, respectively.

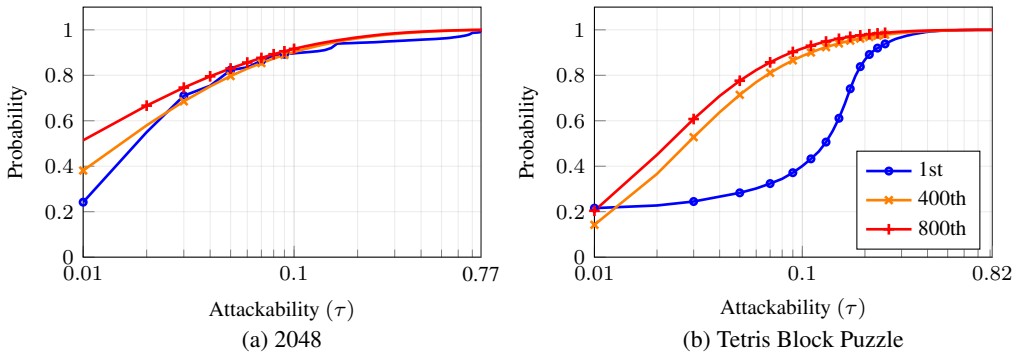

(a) 2048              (b) Tetris Block Puzzle

Figure 15: Cumulative distribution of attackability during training. The minimum $\tau$ is 0, displayed together with 0.01. The maximum $\tau$ is 0.77 and 0.82 in two games, respectively.

## H    DECLARATION ON THE USE OF LARGE LANGUAGE MODELS

The authors recognize the use of OpenAI's ChatGPT and Microsoft's Copilot for proofreading and language refinement of this manuscript. Its contributions were confined to enhancing grammar, style, and clarity, while the technical content remained unchanged.

