# OpenReview forum: "Planning in Stochastic Environments with Awareness of Catastrophic Chance Events"
_ICLR.cc/2026/Conference — Submitted to ICLR 2026_

### Official Review · Reviewer_9RUn · 2025-10-21

**Soundness:** 3
**Presentation:** 2
**Contribution:** 2
**Rating:** 2
**Confidence:** 3

**Summary:**

The paper presents an adapted MCTS algorithm improving an agent's performance in the presence of rare events or adversarial interventions. The key lies in identifying situations, in which the performance is particularly vulnerable to potential perturbations. To this end, the authors introduce the notion of attackability, defined by the product of worst-case performance (expected reward) loss and the likelihood of an intervention. The attackability coefficient is used to inject interventions during MCTS' rollouts. An empirical evaluation in the 2048 and Tetris puzzles demonstrates performance advantages over a baseline MCTS that does not optimize for rare event robustness.

**Strengths:**

Properly dealing with rare events is crucial in many sequential decision-making applications, in particular safety critical applications, where endangering situations are not so common but one needs to be sure that the system reacts to them appropriately. The sparsity of the events combined with their criticality on the agent's performance makes it especially challenging for simulation based RL and MCTS approaches to deal with such problems. The paper presents a new methodology to handle rare event inside MCTS, and they demonstrate empirically that their method can indeed improve performance in two puzzle games.

**Weaknesses:**

The placement of this work in terms of general topic and with respect to related work, and therewith also the impact of the contributions is not clear to me at all. The authors do not properly and formally describe the problem that is addressed. Where do the "catastrophic chance events" come from? Are those events potentially occurring in a stochastic environment, or are those events deliberately injected by an adversarial agent? In the latter case, we are clearly dealing with a two-player game. This, one, raises the question about the rules of this game, and secondly bags for a comparison to other two-player algorithms (which is not provided). If catastrophic events, on the other hand, are stochastic in nature, then handling such problems also has long history in (stochastic) planning and RL -- chance-constrained POMDPs, importance sampling, safe RL, etc. Consequently, the authors should compare their method to such approaches. Despite that the paper contains a related work section, the placement with respect to which is not clear to me at this point.

Despite a clear problem formulation, the clarity of the write up in general leaves much room for improvements. The concept of "afterstate" is not properly introduced, and while knowledge about the general functioning of MCTS can be assumed, the notion of afterstate is rather uncommon. This point is particularly crucial given the centrality of this concept for the authors' overall method. The authors also do not really explain their algorithm changes; they just describe what they change without providing the rationale behind it. The examples assume precise knowledge of the 2048 puzzle game. The paper should hence at least briefly introduce the rules and actions of this game (in the main text).

A discussion of of the effects of the authors' algorithm changes on MCTS' well-known properties (convergence, etc.) is not provided.

Lastly, the benchmark design (2048 and Tetris) is completely artificial, and I can hardly connect them to the general motivation of handling catastrophic events. I would much rather see an evaluation of, e.g., safety critical benchmarks such as the safety gym. In addition to the more clever baselines, all the experiment setup makes it hard for me to judge the actual impact of the developed techniques.

**Questions:**

1. Can you elaborate the specific problem that you want to address (cf. main review)?

2. Why could you not compare to other approaches, like the robust RL method discussed in the paper?

---

> ### Author Response · Authors · 2025-11-23
> **Response to Reviewer 9RUn**
>
> Thank you for sharing your valuable feedback. We have provided answers to each question below.
>
> > The placement of this work in terms of general topic and with respect to related work, and therewith also the impact of the contributions is not clear to me at all
> > Why could you not compare to other approaches, like the robust RL method discussed in the paper?
>
> Please refer to the general response for all reviewers.
>
> > A discussion of of the effects of the authors' algorithm changes on MCTS' well-known properties (convergence, etc.) is not provided
>
> In vanilla MCTS, the UCB formula is known for its asymptotic convergence, which means that the failure probability converges to zero [17]. In AlphaZero, an adapted version of pUCB is used. Though there is no formal proof, it is likely that the convergence properties [18] are still present, given that the predictor follows $p_i > 0$ and $\sum p_i = 1$, which our method guarantees. We will discuss these properties in the context of our method.
>
> [17] Kocsis and Szepesvári. "Bandit Based Monte-Carlo Planning." ECML 2006.\
> [18] Rosin. "Multi-armed bandits with episode context." Annals of Mathematics and Artificial Intelligence 2011.
>
> > the benchmark design (2048 and Tetris) is completely artificial, and I can hardly connect them to the general motivation of handling catastrophic events
>
> Please refer to the general response for all reviewers. We selected these environments because SAZ has been shown to work effectively for them.
>
> > all the experiment setup makes it hard for me to judge the actual impact of the developed techniques
>
> Our experiment setup is designed to demonstrate that **RSZ improves robustness in stochastic environments that are highly sensitive to adversarial interventions**. Specifically, our results demonstrate that RSZ performs well under a very low attack rate of 0.05% (Subsection 4.1), training with this 0.05% attack rate is superior to other larger attack rates (Subsection 4.2), ablation study to show that combining search and attackability performs the best (Subsection 4.3), and an analysis for specific examples (Subsection 4.4). We would greatly apprecitate if reviewer could provide any futher suggestions to improve the design our experiments.
>
> > Can you elaborate the specific problem that you want to address (cf. main review)?
>
> One potential real-world application of our work could be tackling optimization problems, like a job-shop scheduling problem where requests arrive stochastically (but follow a predictable distribution); the agent fails when some tasks are overdue.

---

### Official Review · Reviewer_jTgh · 2025-10-22

**Soundness:** 3
**Presentation:** 3
**Contribution:** 3
**Rating:** 4
**Confidence:** 4

**Summary:**

This paper introduces a stochastic planning approach called Robust Stochastic Zero (RSZ) approach. This approach builds on top of Stochastic Alpha Zero (SAZ) approach. It adds to it by adding robustness to adversarial events — resulting a more conservative plan by the agent and avoidance of most critical actions that can result in game ending states. The authors state that it helps to have the model trained with just the right amount of adversarial actions, if there is over attacking or under attacking in training it can lead to degradation of the performance. The authors claim that their approach achieves 122.1% over the baseline of SAZ in two domains. They perform ablation experiments to show how the configuration they choose is a better one.

**Strengths:**

The overall idea behind the approach of using adversarial attacker to improve the overall robustness of the approach is interesting and definitely relevant to the field.

The authors explain the baseline approach very clearly, which helps with understanding of their overall problem setup.

The paper is quite readable for the most part and is overall well-written. The authors use good amount of formal notations without making the paper dense to read.

**Weaknesses:**

The related work section is not given much thought and is quite underwhelming. It would be useful to keep the related work section in the start of the paper in between introduction and discussion on the preliminary works, as against the end of the paper where there is no surrounding context.

The experiment section is somewhat weak. It raises several questions as against strengthening the original hypothesis of the paper. Until the experiment section, the paper is well written. Thereafter the premise and conclusion of the experiments is not very clear.

The two domains chosen for experimentation tend to have different observations for each experiment. So no observation can be derived conclusively. Maybe it helps to add more domains for experiments to have a more general narrative for each experiment and the corresponding result.

**Questions:**

- For every experiment, clearly state the hypothesis that you start with and how the results support the conclusion of your hypothesis.
- On line 196, there is mention of dynamically determining the attackable threshold. In the experiment setup, How was this value dynamically computed?
- Ablation section currently does not discuss why the other configurations do not yield better results — it just states the scores for each configuration.
- From Figure 5, it seems that when there is no attacker in training and in test, there is best performance. Also, it seems that as attacking increases in training the performances across 4 test groups differs less. It is not clear to me what conclusions we can draw from the different observations available in this figure.
- How was the average performance gain of 122% computed in your experiments?
- Line 170 — typo (c) should be (g)

---

> ### Author Response · Authors · 2025-11-23
> **Response to Reviewer jTgh**
>
> Thank you for sharing your valuable feedback. We have provided answers to each question below.
>
> > The related work section is not given much thought and is quite underwhelming
>
> We will include more related works, discuss different types of environmental stochasticity, and move the section as suggested.
>
> > Thereafter the premise and conclusion of the experiments is not very clear
>
> We aim to improve the robustness of SAZ. Our experiments demonstrate that RSZ delivers improvements with very minimal attack; adding more attacks offers no substantial benefit and can even hurt performance.
>
> > The two domains chosen for experimentation tend to have different observations for each experiment
>
> Minor inconsistencies do not affect our main conclusion. However, we hypothesize that they come from different environmental factors, and we will provide further explanations to clarify them.
>
> > For every experiment, clearly state the hypothesis that you start with and how the results support the conclusion of your hypothesis
>
> We will revise the experiment section to emphasize the objectives and our findings. Thank you for the suggestions.
>
> > On line 196, there is mention of dynamically determining the attackable threshold. In the experiment setup, How was this value dynamically computed?
>
> The threshold is obtained using an online quantile estimator based on previously observed attackability scores. For instance, with a desired rate of 0.05%, we estimate the 99.95% quantile of the attackability score. We detail this in Algorithms 7 and 8 in Appendix B.
>
> > Ablation section currently does not discuss why the other configurations do not yield better results
>
> The ablation shows that the adversarial method with both *tree search* and *attackability measurement* works best. Both removing search (xxx-ns) and attackability (xxx-n$\tau$) at the adversary side yield reduced performance. We will discuss more details in this experiment.
>
> > It is not clear to me what conclusions we can draw from the different observations available in Figure 5
>
> We conclude that RSZ improves performance under very few attacks, as more attacks degrade performance, drawing by comparing $\lambda_\text{train}=0.05\\%$ with other training groups:
> * $\lambda_\text{train}=0.05\\%$ performs better than the baseline ($\lambda_\text{train}=0\\%$), under mostly all $\lambda_\text{test}$ settings.
> * $\lambda_\text{train}=0.05\\%$ also performs better than higher training attack rates ($\lambda_\text{train}>0.05\\%$), under mostly all $\lambda_\text{test}$ settings.
>
> We will present our main findings in a more organized manner.
>
> > How was the average performance gain of 122% computed in your experiments?
>
> $122.1\\%=\frac{122.9\\%+121.3\\%}{2}$ is the average of the performance gains from two environments. In each environment, we first calculate its gain using RSZ's performance divided by SAZ's performance, under the same testing condition $\lambda_\text{test}=0.05\\%$. Then we take the average of the two gains.
>
> > typo (c) should be (g)
>
> Thank you for pointing this out. We will fix it.

---

> > ### Comment · Reviewer_jTgh · 2025-11-25
> > **Acknowledgment of Rebuttal**
> >
> > Thank you for taking time to answer my questions. I read the general rebuttal as well as the specific response to me. Some of my questions regarding related work are now clarified with the explanation in the general rebuttal. But my concerns regarding lack of clarity in the experiments section still stand. The authors responded saying they will clarify the section and add further explanations but did not provide the explanation in response. So it is still not clear to me how the experiment section will improve.

---

> > > ### Author Response · Authors · 2025-11-28
> > > **Response to Reviewer jTgh**
> > >
> > > Thank you for your response to our rebuttal. Below is our further clarification.
> > >
> > > > it is still not clear to me how the experiment section will improve
> > >
> > > All the experiments support that **improving SAZ with rare adversarial attacks is most effective in our target stochastic environments**. We will revise the experiment subsections to emphasize them as follows.
> > > * Sec 4.1, Fig. 3 (0% and 0.05%): The standard RSZ (training in rare attacks) performs well under none and catastrophic settings, supporting our main objective of using rare attacks.
> > > * Sec 4.1, Figs. 3 and 4 (> 0.05%): With more attacks, both performance and stochasticity preservation drop, supporting the importance of preserving stochasticity.
> > > * Sec 4.2, Fig. 5: Training with higher attack rates offers no substantial help, further supporting our point to use rare catastrophes but not aggressive adversarial attacks.
> > > * Sec 4.3, Table 1: The ablation results indicate that combining tree search with attackability is the best, supporting the strength of our method.
> > > * Sec 4.4, Fig. 6: The analysis intuitively shows that the vulnerability decreases during training, supporting our method's effectiveness from a perspective other than performance.
> > >
> > > Also, for minor inconsistencies in the results (e.g., RSZ lower than SAZ in 2048, while not in Tetris), we hypothesize that they are due to game factors and will provide the related discussion.

---

### Official Review · Reviewer_qR9X · 2025-10-31

**Soundness:** 2
**Presentation:** 3
**Contribution:** 2
**Rating:** 4
**Confidence:** 3

**Summary:**

The paper proposes a method for planning in stochastic environments that accounts for the possibility of low-probability events that could, when applied at critical moments, significantly affect performance. The proposed method uses a tree-based planning approach that accounts for an adversary potentially intervening at high-impact times. Results on 2048 and Tetris Block Puzzle show that the proposed method gives better robustness for low attack rates, with little suboptimality in the nominal setting.

**Strengths:**

- The proposed method is likely less conservative than worst-case robust approaches, yet it accounts for low-probability adversaries, effectively finding high-impact attacks through tree-based planning.
- The comparison in Section 4.3 against straighforward methods is a good ablation motivating the proposed method.
- The method's ability to pinpoint the most critical afterstates (Section 4.4) is interesting, as it could potentially also be used to understand how a problem could be slightly modified to obtain a more robust closed-loop system.

**Weaknesses:**

- Missing problem definitions: Section 2 lacks a formal MDP or stochastic game definition, making it difficult to understand what assumptions the method relies on. For instance, is an MDP or POMDP considered? Is a finite state-action space assumed? What is an afterstate? What is a chance event? Defining these terms would help readers familiar with MDPs and POMDPs understand the proposed method.

- Limited related work discussion: The paper does not cite and discuss works in risk-averse (risk-constrained) RL that are relevant and address similar problems. Without this discussion, it is difficult to assess the novelty claims.

- Results: Comparisons against risk-averse or worst-case methods are missing, and would help assess the benefits of the proposed method over existing work. Also, the method is better than SAZ for low attack rates, but it is equivalent or worse for high attack rates, which is surprising as one could expect increased robustness for all attack levels.

- Minor feedback: 1) Acronyms should be defined before they are used, such as "pUCT". 2) In equation (1), there is no need for the symbol $\times$  if it denotes a multiplication. 3) Typos: "corresponds" => "corresponding (line 75), "derivation" => "deviation" (line 276).

**Questions:**

- How does this work positions itself with respect to the literature on risk-averse RL? Adding a paragraph on this topic would strengthen the contribution.
- Please clarify the definition of the severity $T$ in (5) is unclear: $x$ is fixed, so $\min x$ gives $x$, which is probably not the desired definition.
- Results in Figure 3 (b) show worse performance than SAZ for high attack rates, which raises questions regarding the improved robustness of the proposed method. I would have expected improved robustness, even though $\lambda_{\text{train}}<\lambda_{\text{test}}$. Do you have an hypothesis for this performance degradation?
- The concept of critical afterstates and their identification is interesting, (Section 4.4), see my previous comment in "Strengths". Can similar vulnerability identification capabilities be obtained using other methods?

---

> ### Author Response · Authors · 2025-11-23
> **Response to Reviewer qR9X**
>
> Thank you for sharing your valuable feedback. We have provided answers to each question below.
>
> > Missing problem definitions
> > Limited related work discussion
> > How does this work positions itself with respect to the literature on risk-averse RL?
> > Comparisons against risk-averse or worst-case methods are missing
>
> Please refer to the general response for all reviewers. RSZ mostly shares the same concept of risk-averse robust adversarial RL (RARARL), which uses adversarial training to improve robustness while accounting for tail risk. We will revise our manuscript to include a clear problem definition, more related works, and the different types of environmental stochasticity.
>
> > the method is better than SAZ for low attack rates, but it is equivalent or worse for high attack rates, which is surprising as one could expect increased robustness for all attack levels
> > Do you have an hypothesis for this performance degradation?
>
> We hypothesize that RSZ agents not only find a robust strategy but also overfit to it, leading them to perform significantly better than the baseline at low attack rates but worse at high attack rates. However, assuming that catastrophes are normally rare, this is not a major issue.
>
> > Minor feedback
>
> We will fix the minor issues accordingly.
>
> > Please clarify the definition of the severity $T$ in (5) is unclear
>
> In equation (5), $x = \\{q_1, ..., q_n\\}$ is the set of mean values of all $n$ chance events, $\min x$ is the minimum among them. We will use bold text to distinguish vectors (e.g., $\boldsymbol{p}$, $\boldsymbol{\rho}$, $\boldsymbol{x}$, $\boldsymbol{\delta}$) for better clarity.
>
> > Can similar vulnerability identification capabilities be obtained using other methods?
>
> We leverage the concept of critical states that has been applied in RL research, but extend it to critical afterstates. Previous designs cannot exactly meet our requirements for rare attack scenarios. They mostly consider only state values or outcomes. Instead, our metric further considers the distribution of all chance events' values to prioritize rare cases.

---

> > ### Comment · Reviewer_qR9X · 2025-11-25
> >
> > Thank you for your replies and the general response to the reviews, which I carefully read. In particular, I appreciate that the authors plan to revise the manuscript to include a clearer problem definition, more related works, and the different types of environmental stochasticity. However, since these revisions appear to be substantial and they are not yet provided, I will maintain my current score.

---

### Official Review · Reviewer_342r · 2025-11-01

**Soundness:** 3
**Presentation:** 3
**Contribution:** 3
**Rating:** 6
**Confidence:** 4

**Summary:**

The paper proposes Robust Stochastic Zero (RSZ), a planning framework for stochastic environments that aims to be robust to rare but catastrophic chance events without distorting the environment's inherent randomness. Instead of the stepwise minimax adversaries common in robust RL, RSZ introduces a lurking adversary that selectively intervenes only at critical afterstates, i.e., chance nodes where some outcomes would trigger avoidable catastrophes. Criticality is quantified by an attackability score that combines severity and rarity. RSZ integrates this notion into a modified search procedure, Robust Stochastic MCTS, layered on Stochastic AlphaZero/MuZero, and augments the network with a head that predicts value‑drop magnitudes for chance outcomes to guide selective interventions. In experiments on 2048 and Tetris Block Puzzle, agents trained with a tiny attack rate outperform baselines under equally rare attacks, remain comparable with no attacks, and preserve empirical tests of environment randomness at small attack rates.

**Strengths:**

(1) The paper articulates a nice gap i.e., expectation‑centric planning tends to ignore tail events, while stepwise minimax adversaries distort dynamics and induce over‑conservatism. The lurking adversary that intervenes rarely is a compelling middle ground.

(2) Focusing criticality at afterstates is in my opinion a good design choice for stochastic MCTS (it isolates the locus where nature acts and aligns with SAZ/SMZ's separation of decision and chance nodes).

(3) Decomposing criticality into (i) severity and (ii) rarity is an intuitive, implementable criterion. The normalization choices (tanh, max‑normalization) aim to make the score stable across scales and arities.

(4) The empirical results are very promising.

**Weaknesses:**

(1) The work would be stronger if it situated RSZ within a known formal objective (e.g., a budget‑limited adversary model, distributionally robust return, or a tail‑risk criterion). As written, \tau and the thresholding policy are well‑motivated but heuristic. This leaves questions about what RSZ is optimizing (e.g., an upper bound on tail risk, a CVaR‑like surrogate under a budget constraint, etc.) and about convergence properties of RS‑MCTS under that objective.

(2) There is no argument that, with accurate \delta and an appropriate \tau_th estimator, RS‑MCTS improves tail risk, bounds regret against a \lambda‑budget adversary, or preserves expectation in the no‑attack limit. Even a simple proposition (e.g., monotonic effect of \lamda on a tail metric) would help.

(3) (a) The claim under equation (6) i.e., "falls short of the parent value q..." is not necessarily true. Because softmax(a + const) = softmax(a), subtracting q has no effect. \delta depends only on the relative ordering and spread of the x's. That is fine in practice, but the text should not claim it is directly tied to q; while it clearly isn't.

(3) (b) \sigma tending to 0 (for example, if all event values are (near) equal) would lead to ill-conditioned values, I suppose the text should infact be refactored to represent a principled \epsilon stabilization.

(4) (a) I believe the novelty claim should be contextualized i.e., "First method to be aware of catastrophes while maintaining inherent stochastic dynamics" is too strong. There are established strands of risk‑sensitive and robust planning that try to protect tails without fully adversarializing every step (e.g., tail‑risk criteria, constrained risk, distributionally robust value backups). It is fine to to not cite specific works here, but the claim should acknowledge neighboring paradigms and be scoped precisely.

(4) (b) The coverage of the related works section is too narrow.

(5) In SMZ, how do you prevent attacked events from corrupting supervision of \rho (and r)? Do you mask attacked samples when training \rho, or correct them with importance weights?

(6) The core claims are broad (robust planning in stochastic environments). But the evidence (from expts) comes from 2 single‑player puzzles. Ablations and Table 1 are shown for only one of them. It would strengthen the case to replicate key ablations in Tetris Block Puzzle.

(7) From my understanding, tests that aggregate over all steps (e.g., overall position frequency uniform over empty cells) can be easily confounded by which states the agents visit. Importantly, RSZ may steer into different state distributions. A deviation in position frequency may reflect state visitation changes, not chance‑rule bias.  I would suggest conditioning the tests on state context: e.g., compute uniformity per step over the set of empty cells and aggregate step‑wise p‑values (or use weighted tests that normalize by the number of empty positions per step). Similarly, sequence tests should control for non‑IID structure induced by varying legal positions/events across states.

Other comments (parts that I might not have fully understood):

(8) The paper's narrative emphasizes low‑probability catastrophes, but does H (Eq. 6) not measure outlierness among child values? i.e., not the probability of those children under \rho. As a result, can \tau not be large even when the catastrophic child is not rare?

(9) Looks like T and H are sensitive to the scale and dispersion of child values. But there is no demonstration that \tau is affine‑invariant or robust across tasks with different reward scales, which complicates cross‑domain tuning and the \tau threshold estimator.

(10) If adversarially chosen chance events are treated as observed during training, won't they bias the learned \rho (and potentially r)? i.e., I am talking about potential contamination of model heads.

(11) Minor: the work claims RS‑MCTS "ensures" every chance outcome at root is visited at least once, but this may be budget‑infeasible (right?) in high‑arity states. There is no fallback rule for incomplete vectors and no accounting of the opportunity cost (reduced depth).

**Questions:**

Please see the weaknesses section.

---

> ### Author Response · Authors · 2025-11-23
> **Response to Reviewer 342r**
>
> Thank you for sharing your valuable feedback. We have provided answers to each question below.
>
> > The work would be stronger if it situated RSZ within a known formal objective
>
> RSZ mostly shares the same concept of *risk-averse robust adversarial RL* (RARARL), which uses adversarial training to improve robustness while accounting for tail risk. We will improve the manuscript by including the related objectives and discussing the convergence.
>
> > There is no argument that, ..., RS‑MCTS improves tail risk, bounds regret against a \lambda‑budget adversary, or preserves expectation in the no‑attack limit.
>
> We only empirically demonstrate that the tail performance is improved, the vulnerability of afterstates is reduced, and the expectation is preserved. We will consider adding theoretical analysis in the future. Thank you for your suggestion.
>
> > The claim under equation (6) i.e., "falls short of the parent value $q$..." is not necessarily true
>
> We originally denote equation (6) in this way to make $T$ and $H$ more consistent. However, we will fix the problematic descriptions while keeping our intention, and consider changing the term "value drop magnitudes" accordingly.
>
> > $\sigma$ tending to 0 (for example, if all event values are (near) equal) would lead to ill-conditioned values
>
> We will clarify that we use a stabilization $\sigma + 10^{-8}$ in our implementation.
>
> > I believe the novelty claim should be contextualized
> > The coverage of the related works section is too narrow
>
> For clarification on our target environment, please refer to the general response for all reviewers. We will include more related works and discuss different types of environmental stochasticity.
>
> > In SMZ, how do you prevent attacked events from corrupting supervision of \rho
>
> The attack events can corrupt $\rho$, but we consider this acceptable. As the attack rate is very low, most of the learned probabilities remain unaffected. If $\rho$ for a vulnerable afterstate is tied to an attack event, it actually helps the agent become better aware of the rare catastrophes.
>
> > It would strengthen the case to replicate key ablations in Tetris Block Puzzle
>
> Due to limited time and computation resources, the ablations in Tetris Block Puzzle are not provided in the submission. We will take them into account in the future.
>
> > tests that aggregate over all steps (e.g., overall position frequency uniform over empty cells) can be easily confounded by which states the agents visit
>
> We do aggregate p-values over all steps, but before doing so, they are calculated separately with considering the afterstate conditions:
> * For the position frequency, we categorize afterstates based on the shapes of empty cells, and the p-value for each set is then calculated separately, e.g., Figure 12 (e) and (h) are in the same set; their new tile should be placed uniformly among four spaces.
> * For the sequence test, we extract only the tile type for calculation, e.g., a sequence of 2, 2, 4, 2, … in 2048. The tile location or current state configuration is not taken into account.
>
> > As a result, can \tau not be large even when the catastrophic child is not rare?
>
> In our design, $H$ considers only chance events' values. That is, it could sometimes target low-value, high-probability cases, e.g., a two-event case with $x = \\{10, 100\\}$ and $\rho = \\{0.9, 0.1\\}$. However, this is acceptable as high-probability events are commonly sampled by the environment anyway, so we do not also consider $\rho$ in $H$ for simplicity.
>
> > no demonstration that \tau is affine‑invariant or robust across tasks with different reward scales
>
> $T$ and $H$ will work in environments with different value/reward scales. The normalizations in equations (5) and (6) ensure $\tau \in [0,1]$. The estimator is an online quantile estimator that adaptively fits in different $\tau$ distributions. Note that 2048 and Tetris Block Puzzle have different reward scales, with the former reaching up to thousands and the latter only in the tens. As demonstrated by our experiments conducted using the same hyperparameters, cross-domain tuning is not necessary.
>
> > this may be budget‑infeasible (right?) in high‑arity states
>
> For environments where enumerating all chance events is infeasible, a potential fallback can be the *completed Q-values* [15], which takes the parent afterstate value to fill unvisited chance events. Furthermore, to better support continuous chance event spaces, *Sample MuZero* [16] is another promising direction.
>
> [15] Danihelka et al. "Policy improvement by planning with Gumbel." ICLR 2022.\
> [16] Hubert et al. "Learning and planning in complex action spaces." ICML 2021.

---

### Official Review · Reviewer_Vm4N · 2025-11-04

**Soundness:** 3
**Presentation:** 2
**Contribution:** 2
**Rating:** 2
**Confidence:** 3

**Summary:**

The paper considers the problem of performing robust planning in stochastic environments that are subject  to rare, catastrophic events. A specific model for such stochastic environments with rare, catastrophic events is proposed. For this model, a version of stochastic Monte Carlo tree search (MCTS) that leverages a neural network to estimate the system model, called Robust Stochastic Zero (RSZ), is proposed as a solution method. The core difference between this solution method and existing MCTS approaches is a procedure for inflating the perceived probability of the outcome of the rare, catastrophic events during the backpropagation phase of MCTS. Experiments are provided comparing the performance of RSZ with Stochastic Alpha Zero (SAZ), a practical implementation of non-robust MCTS for stochastic environments, on the games 2048 and Tetris.

**Strengths:**

Development of robust learning methods for sequential decision-making problems is an extremely active area, and the goal of developing versions of MCTS for stochastic environments that incorporate robustness to high-impact, low-probability events is a worthy one. For this reason, the topic of the paper is timely and likely of interest to the community. In addition, the technical presentation of the proposed model and method is solid, supporting reproducibility.

**Weaknesses:**

1. The paper suffers from clarity issues. Specifically, insufficient explanation is provided of the reasoning behind key steps in the proposed model and proposed approach. Importantly, the rationales underlying the notion of attackability in Sec. 3.2 and equations (5) and (6) and the procedure in  equations (7) and  (8) for inflating the perceived outcomes of rare, catastrophic events are unclear. This makes it difficult to connect the proposed model and approach to concrete applications, and it also makes it challenging to accurately assess how they differ from existing models and methods.
2. The motivation behind the specific model proposed in Sec. 3.1 and Sec. 3.2 for stochastic environments with rare, catastrophic events is weak. Specifically, it is not clear what kinds of real-world applications or important problems with relevance to the research community are captured by this model. This makes it difficult to accurately assess the significance of the proposed model. It would be useful to provide specific examples of important problems that either fit this model or where the model and corresponding approach lead to good approximate solutions.
3. The proposed approach, RSZ, appears to be designed to  apply only to problems that can be captured using the specific model mentioned in Weakness 2 above. In particular, the mechanism described in Sec. 3.3 and in equations  (7)-(8) for inflating the perceived outcome probabilities for rare, catastrophic events are tied to the notions of attackability and the risk thresholds that are developed in Sec. 3.2. This dependence makes it unclear whether the proposed approach can apply to problems beyond the specific model the authors propose.
4. There are two main issues with the experimental results:

    (i) Though the abstract and introduction present the paper as overcoming the overly  conservative nature of existing robust planning and robust learning methods, the only baseline that the experiments compare against is SAZ, which is a non-robust MCTS method for stochastic problems. This makes is difficult to situate the proposed method within the very active literature on methods for robust learning in stochastic environments, an overview of which is provided in the related works in Sec. 5. Comparison with key representatives of the robust methods described in the related works is needed.

    (ii) There are inconsistencies in the experimental results presented. Specifically, in Fig. 3, RSZ trained with $\lambda_{train} = 0.05\%$ is shown to mostly outperform SAZ (which is equivalent to RSZ with $\lambda_{train} = 0\%$) for a variety  of values of $\lambda_{test}$. However, Fig. 5 appears to show that SAZ ($\lambda_{train} = 0\%$) outperforms RSZ for all values of $\lambda_{train} >  0\%$ on both environments, and Figs. 7, 9, 10, and 11 in the appendix show decidedly mixed results between the two approaches. These issues need to be clarified to enable accurate assessment of the effectiveness of RSZ.

**Questions:**

1. What is the rationale behind the selection rules in equations (1) and (2)?
2. What are the precise definitions of $p^\ell, v^\ell, \rho^\ell, \nu^\ell$ in line 88? How are they going to be obtained?
3. What is the reasoning behind the update rule in equation (3)?
4. What is the intuitive reasoning behind the notion of "attackability" defined in Sec. 3.2? Why is the definition provided a good one?
5. What is the rationale behind equations (4), (5), and (6)? Are there alternatives, and why is this definition preferable to them?
6. Regarding the "Attackable Threshold" defined in lines 190-198: what concrete types of scenarios can this model capture and how?
7. What is the intuition and rationale behind the specific form of the update given in equations (7), (8)?
8. Regarding the **Decision** description on lines 264-269: for what specific scenarios is this type of adversary decision procedure a good model?
9. Can you comment on the inconsistencies mentioned in Weakness 4(ii) from **Weaknesses** above?
10. Is there a mitigating reason that no robust planning or learning baselines were compared with in the experiments?

---

> ### Author Response · Authors · 2025-11-23
> **Response to Reviewer Vm4N**
>
> Thank you for sharing your valuable feedback. We have provided answers to each question below.
>
> > The paper suffers from clarity issues
>
> We will improve the clarity by adding more explanations.
>
> > it is not clear what kinds of real-world applications or important problems with relevance to the research community are captured by this model
>
> One of the potential real-world applications is *optimization problems*. For example, a job-shop scheduling problem where requests arrive stochastically (but follow a predictable distribution); the agent fails when some tasks are overdue. However, as our main goal is to improve SAZ, we select the same benchmarks used in SAZ for better comparison.
>
> > RSZ, appears to be designed to apply only to problems that can be captured using the specific model
>
> We agree with the reviewer that this is specifically designed for this kind of problem. However, while SAZ represents the SOTA in this domain, extending SAZ with robustness brings significant novelty.
>
> > Comparison with key representatives of the robust methods described in the related works is needed
>
> Please refer to the general response for all reviewers.
>
> > inconsistencies in the experimental results presented
>
> Fig. 5 reorganizes statistics from Fig. 3, not an inconsistency, e.g., Fig. 3 (a) left-most 1st, 2nd, 3rd, 4th bins become Fig. 5 (a) 1st, 5th, 2nd, 6th bins, respectively. Figs. 10 and 11 show the same conclusion that RSZ generally improves tail performance at the expense of a slight loss in top performance.
>
> For other inconsistencies, e.g., Fig. 7, we attribute them to RSZ agents in 2048 learn more slowly. Specifically, when a large tile is about to be created, only a few empty spaces left. A catastrophe is likely to exist and can be exploited, e.g., Fig. 12 (a)-(d), making agents learn slowly as they cannot easily find robust strategies to create larger tiles. On the other hand, no such characteristic in Tetris Block Puzzle. We will add explanations to clarify the results.
>
> > What is the rationale behind the selection rules in equations (1) and (2)?
> > What are the precise definitions of $p^\ell$, $v^\ell$, $\rho^\ell$, $\nu^\ell$ in line 88?
> > What is the reasoning behind the update rule in equation (3)?
>
> These background techniques are used by the original SMZ. Due to the page limit, we cannot provide all details in the main text.
> * Equation (1) is applied to players' action selection, including exploitation (the first term) and exploration (the second term, diminishing when $n_a$ increases). Equation (2) is designed for environments' stochastic chance events (consider true probability $\rho_c$).
> * During the MCTS expansion phase, SMZ obtains them using a neural network. $p^\ell$ is the prior probabilities for taking all actions. $v^\ell$ is the value of $s^\ell$. $\rho^\ell$ is the environment probabilities for all chance events. $\nu^\ell$ is the afterstate value of ${as}^\ell$.
> * Equation (3) updates the mean value $q^i$ using $G^i$. Rewards between $i$ and $\ell$ are additionally added in $G^i$ since $v^\ell$ only covers those after $\ell$.
>
> > What is the intuitive reasoning behind the notion of "attackability"
>
> The attackability is designed to identify chance events whose values are significantly worse than others, where we separately measure *how bad it is* and *how it compares with others* using equations (5) and (6):
> * In equation (5), the core is $-\frac{\min{x}}{q}$, the ratio of the worst-case event value and its parent value; the other part is a transformation. It gets a higher result when the difference is larger.
> * In equation (6), the share of max in $\delta$ is measured. It gets a higher result when $\delta$ is closer to a one-hot, namely one value is significantly worse than others.
>
> The current equations are one of the approach reflecting the measurement concept. There can be other designs.
>
> > Regarding the "Attackable Threshold" ... what concrete types of scenarios can this model capture and how?
>
> The attackable threshold identifies the most vulnerable afterstates that the adversary may exploit. Such a concept is applicable to scenarios where each chance events have a corresponding value indicator (which can normally be obtained using a neural network).
>
> > What is the intuition and rationale behind the specific form of the update given in equations (7), (8)?
>
> Intuitively, for non-attack node in equation (8), $q^i$ is computed by approximately averaging values of all its children ($\sum_a …$). In contrast, for the attack node in equation (7), as the $i$th node is determined to being attacked, $q^i$ is computed by considering only the value of attacked child, so there are no $\sum$.
>
> > what specific scenarios is this type of adversary decision procedure a good model?
>
> It only attacks in the most vulnerable situations while keeping less vulnerable ones untouched, which is designed for addressing catastrophes in stochastic environments where aggressive interventions degrade average performance.

---

> > ### Comment · Reviewer_Vm4N · 2025-11-25
> >
> > Thanks to the authors for their response. I continue to have the primary concerns that I outlined in the four points in the **Weaknesses** section.
> >
> > Regarding **W1**: while the additional discussion has somewhat clarified equations (5) and (6), the rationale behind the outcome probability inflation procedure in equations (7)-(8) remains unaddressed.
> >
> > Regarding **W2**: no concrete examples are given of real-world problems or theoretical problems of interest to the community that are captured by the model proposed in Sec. 3.2-3.3; this remains a critical weakness of the work.
> >
> > Regarding **W3**: your response seems to suggest that your work provides a robust extension to SAZ for general problems; in fact, the scope of this work is narrower, as the proposed method only applies to problems of the very specific form described in Sec. 3.2-3.3. Significant additional clarification regarding why this specific model is useful and applicable is needed in order to substantiate the significance of the proposed method.
> >
> > Regarding **W4**: the abstract and introduction situate the proposed method as overcoming shortcomings of other robust methods, yet the only baseline compared against is the non-robust SAZ method.

---

> > > ### Author Response · Authors · 2025-11-28
> > > **Response to Reviewer Vm4N**
> > >
> > > Thank you for your response to our rebuttal. Below are further clarifications regarding the concerns you raised.
> > >
> > > > the rationale behind the outcome probability inflation procedure in equations (7)-(8) remains unaddressed
> > >
> > > The rationale behind this design is to emphasize the worst-case events that are likely to be ignored in the normal procedure. To efficiently propagate the worst-case attack event, we use equation (7) to keep the attack event and discard other non-attack events. Otherwise, we keep the mean value updated normally using equation (8).
> > >
> > > For example, a mean value $q = 10$ is calculated by 10 visits during MCTS. At the 11th visit, assuming a catastrophe with a mean value of 5 is exploited. The inflation procedure (7) will set $q \approx 5.5$, while the normal procedure (8) will keep $q \approx 9.5$. A lower $q$ can better let the player be aware that it is risky.
> > >
> > > > no concrete examples are given of real-world problems
> > >
> > > The related real-world problems are *optimization problems*, such as the job-shop scheduling problem discussed in our previous response. We will revise the paper to indicate this point.
> > >
> > > > the scope of this work is narrower
> > >
> > > Thank you for pointing out this concern. We do not think the scope is narrow, as *robust extension of AlphaZero- and MuZero-based algorithms* is a promising research direction. For example, both [19] and [20] are recent works aimed at improving the robustness of this family of algorithms, within different scopes than ours. We will highlight the significance in our paper.
> > >
> > > [19] Moss et al. "ConstrainedZero: Chance-Constrained POMDP Planning using Learned Probabilistic Failure Surrogates and Adaptive Safety Constraints." IJCAI 2024.\
> > > [20] Li et al. "RobustZero: Enhancing MuZero Reinforcement Learning Robustness to State Perturbations." ICML 2025.
> > >
> > > > the abstract and introduction situate the proposed method as overcoming shortcomings of other robust methods
> > >
> > > We do not intend to overcome the shortcomings of other robust methods, but rather to explain **why we cannot leverage a similar aggressive adversary in these environments**. This forms the core of our approach: minimizing attacks while preserving the dynamics. We'll update the abstract and introduction to make our target clearer and avoid confusion.

---

### Author Response · Authors · 2025-11-23
**General Response**

Dear reviewers,

Thanks for taking the time to review our paper and for sharing your valuable comments and feedback. We truly appreciate it.

We'll start by addressing the general questions raised by reviewers regarding the *stochastic environment* and the *difference between the scope of our work and previous works*. Responses to each reviewer's specific points are provided under your individual review forms.

We hope our clarifications address the reviewer's concerns and strengthen the contributions of our work. Please feel free to let us know if you have any other suggestions or further questions.

---

### **Clarification of the stochastic environment**

Stochastic environments come in various types, each with its own unique characteristics.
1. Stochasticity arises **in the same way as actions do**, such as in Stochastic MuJoCo, where an agent applies a physical force, and then the environment adds another random physical force.
2. Stochasticity arises **differently than actions do**, such as in the 2048 game, where an agent slides the puzzle, and then the environment adds a new random tile.
3. Stochasticity arises **in rewards**. Namely, the reward obtained from state transitions is stochastic.

In this work, we focus on **type 2**, modeled using afterstate and chance events. To our best, related works mostly focused on type 1 [1]-[7] or type 3 [8]-[10]. Their approaches are not tailored to our target, e.g., action perturbations will not work in 2048, as the stochasticity comes differently than action. While some [8], [11] have also explored type 2, their tests were conducted on small-scale problems with a maximum episode length of less than 100. In contrast, we target more challenging problems where the episode length can be thousands or more.

To avoid confusion, we will improve our manuscript, clarifying our target and including necessary related works with a comprehensive comparison. In addition, *we note that our target stochastic environments have not been widely studied in this field, which is precisely what makes our work novel*.

[1] Chow et al. "Risk-sensitive and robust decision-making: a cvar optimization approach." NeurIPS 2015.\
[2] Chow et al. "A lyapunov-based approach to safe reinforcement learning." NeurIPS 2018.\
[3] Eysenbach et al. "Maximum entropy RL (provably) solves some robust RL problems." ICLR 2022.\
[4] Li et al. "RobustZero: Enhancing MuZero Reinforcement Learning Robustness to State Perturbations." ICML 2025.\
[5] Mandlekar et al. "Adversarially robust policy learning: Active construction of physically-plausible perturbations." IROS 2017.\
[6] Pan et al. "Risk averse robust adversarial reinforcement learning." ICRA 2019.\
[7] Rigter et al. "One risk to rule them all: A risk-sensitive perspective on model-based offline reinforcement learning." NeurIPS 2023.\
[8] Greenberg et al. "Efficient risk-averse reinforcement learning." NeurIPS 2022.\
[9] NS et al. "Extreme risk mitigation in reinforcement learning using extreme value theory." TMLR 2023.\
[10] Zhou et al. "Is Risk-Sensitive Reinforcement Learning Properly Resolved?" arXiv:2307.00547 2023.\
[11] Chow et al. "Risk-constrained reinforcement learning with percentile risk criteria." JMLR 2018.

---

### **Why SAZ and why not compare with other methods?**

Our scope is to enhance the SAZ algorithm, which is the SOTA for a particular type of stochastic environment. Planning in such environments requires non-trivial considerations due to their stochastic transitions and long trajectories, such as afterstate and tree search [12]-[14]. **To the best of our knowledge, our work is the first attempt to address robustness under these challenges, highlighting our novelty.** Moreover, these challenges make it difficult to directly apply related works in our domain (e.g., action perturbations cannot be applied; non-search-based methods generally do not perform as well as search-based ones). Therefore, to clearly demonstrate the improvement within our scope, a direct comparison between our method and SAZ is most appropriate. However, if reviewers feel certain related works should be compared, please let us know, and we are willing to include them.

[12] Antonoglou et al. "Planning in stochastic environments with a learned model." ICLR 2021.\
[13] Szubert and Jaśkowski. "Temporal difference learning of n-tuple networks for the game 2048." IEEE CIG 2014.\
[14] Yeh et al. "Multistage temporal difference learning for 2048-like games." IEEE TCIAIG 2016.

---

### Meta-Review · Area_Chair_MUyx · 2026-01-05

**Summary:**

The paper proposes a planning method for stochastic environments that aims to improve robustness against rare but catastrophic chance events by selectively intervening at critical afterstates.

Strength: Reviewers generally agreed that the idea of focusing on rare catastrophic events while avoiding overly conservative adversarial interventions is interesting and timely.
Weaknesses:
1. Clarity: The primary weakness of the work is a lack of clarity. This issue was explicitly highlighted by three reviewers (Vm4N, jTgh, and 9RUn).
2. Evaluation: The most significant weakness concerns how the method is evaluated. Reviewer Vm4N explicitly suggested that the authors should include direct comparisons with other robust methods. Similarly, reviewer qR9X pointed out that comparisons against risk-averse or worst-case baselines are missing. Although the authors explained why they focused on comparisons with SAZ, the absence of such baselines makes it difficult to accurately assess the significance of the proposed method.

**Reviewer Concerns:**

Reviews believe that the primary weakness of the paper is a lack of clarify and limited comparison besides SAZ.
During the rebuttal, authors mentioned that the paper improves the SOTA method, SAZ. Authors also clarify that the problem addressed is the second one (type two).

1. Stochasticity arises in the same way as actions do, such as in Stochastic MuJoCo, where an agent applies a physical force, and then the environment adds another random physical force.
2. Stochasticity arises differently than actions do, such as in the 2048 game, where an agent slides the puzzle, and then the environment adds a new random tile.
3. Stochasticity arises in rewards. Namely, the reward obtained from state transitions is stochastic.

However, even with the explanation, reviews may still point out the clarify issue and limited contribution.

**Reviewer Scores:**

It will not be changed much.

---

### Decision · Program_Chairs · 2026-01-26

Reject